# Temporal Instability and Transferability Analysis of Daytime and Nighttime Motorcyclist-Injury Severities Considering Unobserved Heterogeneity of Data

**Chamroeun Se** [1], **Thanapong Champahom** [2], **Sajjakaj Jomnonkwao** [3], **Panuwat Wisutwattanasak** [1], **Wimon Laphrom** [1] and **Vatanavongs Ratanavaraha** [3,*]

1   Institute of Research and Development, Suranaree University of Technology, Nakhon Ratchasima 30000, Thailand
2   Department of Management, Faculty of Business Administration, Rajamangala University of Technology Isan, Nakhon Ratchasima 30000, Thailand
3   School of Transportation Engineering, Institute of Engineering, Suranaree University of Technology, Nakhon Ratchasima 30000, Thailand
*   Correspondence: vatanavongs@g.sut.ac.th

**Abstract:** Using motorcycle crash data from 2016 to 2019, this paper aims to uncover and compare the risk factors that influence the severity of motorcyclist injuries sustained in daytime and nighttime motorcycle crashes in Thailand. Mixed-ordered probit models with means and variances in heterogeneity were used to take into consideration unobserved heterogeneity. The temporal instability of risk factors was also extensively explored. The results show that male motorcyclists, speeding, fatigue, crashes in work zones, crashes on raised median roads, intersection-related crashes, crashes on wet roads, and crashes on unlit roads are all factors that are positively associated with the risk of death and serious injury in nighttime crashes. The presence of pillions, crashes on two-lane roads, crashes on depressed/flush median roads, crashes in rural areas, U-turn-related crashes, weekend crashes involving heavy vehicles, and head-on crashes are factors that were positively associated with risk of death and serious injury for both daytime and nighttime crashes. This study's findings provide evidence that factors that influence motorcycle accidents during the daytime and nighttime vary significantly. Additionally, nighttime crashes typically carried a higher risk of fatalities or serious injuries compared to daytime crashes. A discussion of policy recommendations is also provided.

**Keywords:** death and serious injuries; developing country; middle-income country; mixed ordered probit; random parameters; heterogeneity in means and variances

## 1. Introduction

Road accidents are a quiet epidemic stalking humanity. Globally, road and traffic crashes are among the most dangerous and common risks, resulting in 1 fatality every 24 s [1]. Every year, approximately 1.35 million fatalities and 50 million catastrophic injuries occur [2] and have severe consequences including loss of human resources and untold or unknown suffering for the families who must deal with grief or crippled relatives [3–6]. Traffic accidents can cause entire families to fall into poverty by taking the breadwinner out of the equation or by costing money for lost wages and extended medical treatment. Nine out of ten victims reside in low- and middle-income countries (LMIC) [1]. Developing countries lose between 2% and 5% of their GDP each year as a result of fatalities and catastrophic accidents on the road [1]. In the Southeast Asia region alone, deaths resulting from road traffic accidents produce approximately 316,000 victims per year, and vulnerable road users such as pedestrians, cyclists, and especially motorcyclists account for 50% of this number [1].

In Thailand, for example, motorcycle accidents make up the equivalent of more than 30% of all annual non-motorcycle accidents (Figure 1). Additionally, according to a report

by the World Health Organization, the estimated cost of death and serious injuries due to road accidents was $44.71 billion which is equivalent to 10.9% of the country's GDP in 2016 [2]. Moreover, compared to other road users' death rates, motorcyclist fatalities contributed to 74% of all deaths due to road accidents, followed by 8% for pedestrians, 6% for passengers of 4-wheeled cars, and 6% for drivers of 4-wheeled cars (as shown in Figure 2). Therefore, it is imperative to investigate and completely comprehend the factors that contribute to the seriousness of motorcycle crashes.

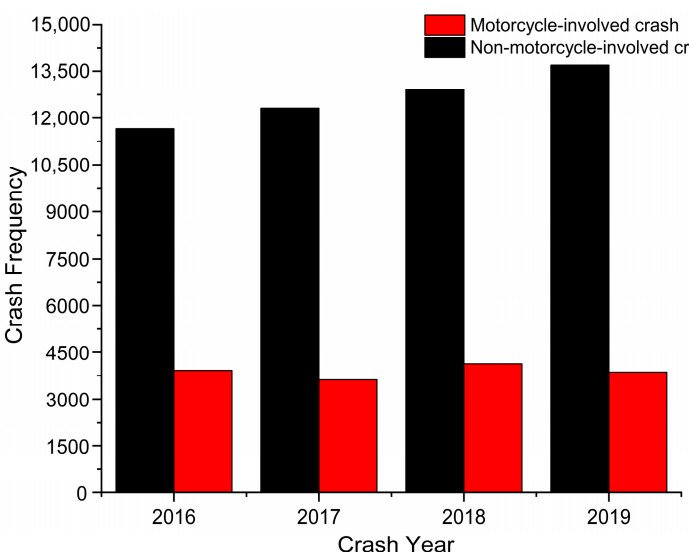

**Figure 1.** A comparison of crash frequency between motorcycle-involved crashes versus non-motorcycle-involved crashes in Thailand in 2016–2019, based on crash data from the Department of Highways (DOH).

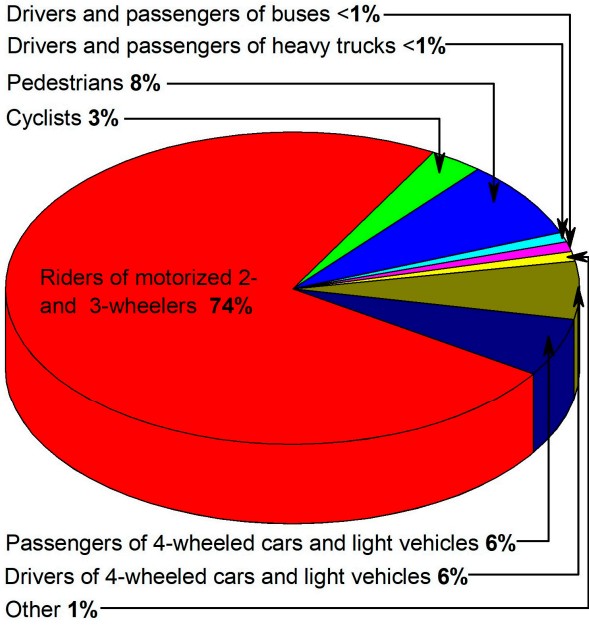

**Figure 2.** 2016, Injury surveillance system, Bureau of Epidemiology, Department of Disease Control, Ministry of Public Health (Source: adapted from Global status report on road safety 2018, WHO).

With regard to the time of day of the crashes, Behnood and Mannering [7] cited two explanations for the diversity of the contributing factors to injury severity. First, human behavior such as decision-making, responsiveness, alertness, etc., might change during the

day because of weariness, biorhythm, lack of sleep, etc. Second, the unobserved variables affecting sight and brightness may change depending on the time of the day (during the day and the night). Given this, various collision severity studies have carefully considered the impact of the time of day on the severity of injuries. Studies have covered topics including the injury severity of highway-rail grade-crossing crashes [8], heavy-truck crashes [7], pedestrian-involved crashes [9–11], hilly expressway crashes [12], the severity of work zone crashes [13], and the severity of bicycle-vehicle crashes [14]. Empirical evidence from these studies suggests that the time of the day is significant in determining the severity of the injuries resulting from a collision and that this influence may extend beyond the basic use of indicator variables (showing various time-of-day intervals) in statistical models [7].

Another new problem in accident injury investigations is temporally shifting model parameters. Human behavior is ever-evolving for a variety of reasons, including natural evolution, technological development (not just transportation but also societal effects), and long-term structural shifts brought on by continuously evolving behaviors, and these changes are likely to affect the outcome of injuries over time [15]. Behnood and Mannering [16] discovered that the urban nature of crash data, changes in police-reporting practices, advancements in vehicle safety features and drivers' reactions to them, and effects of macroeconomics significantly affected the temporal instability of factors influencing the severity of drivers' injuries in single-vehicle crashes. Similar to this, in their motorcycle crash studies, Alnawmasi and Mannering [17] found temporal instability in the effect of factors determining motorcyclist injury outcomes and identified some potential sources including changes in rider skills and behavior, motorcycle performance and technology, and changes that are brought on by how motorcyclists adapt to the shifting behavior and attitude of other vehicle users who may also have their behavior shifted or changed due to evolving use of personal technologies in their vehicles over time and technological innovations such as new vehicle technologies that may fundamentally change the way that humans drive vehicles. These alterations in travel habits and attitudes throughout time are likely to have a temporal impact on the outcomes of ensuing crashes. Therefore, the assumption that the effects of the contributing elements on rider injury severities will be constant over time seems very unlikely, and its statistical estimation results could result in incorrect conclusions and inefficient or unsafe safety measures [15].

However, according to literature studies, there is a research vacuum in the assessment of the variations between daytime and nighttime contributing elements of rider injury severities and examining how those contributing factors have evolved over time. Therefore, the purpose of this study is to close the gap by offering thorough responses to the following important questions:

- Are the causes of the severity of motorcyclists' injuries in daytime incidents different from those of nighttime crashes? How are they different?
- Are the contributing elements to the seriousness of daytime and nighttime motorcyclists' injuries temporally stable?

Although previous research has considered temporal instability in examining the severity of injuries sustained by motorcyclists [17–19], this current study stands out because it is unique in its approach to investigating the temporal instability of the factors that contribute to injury severities, using different perspectives than those employed in earlier studies. That is, first, the study utilizes data from Thailand, a developing country where motorcycles make up the majority of registered vehicles, and it employs various sets of observations and additional explanatory variables, which may have specific parameter distributions due to varying levels of motorcyclist injury severity. Second, the study examines the discrepancies between daytime and nighttime motorcyclist injury severities while also considering temporal instability and unobserved heterogeneity. Through non-transferability and temporal instability, the findings of the current study may contribute to the existing literature by providing valuable knowledge for practitioners, researchers, institutions, and decision-makers to improve highway safety, particularly concerning

motorcyclist safety, and facilitate the development of more efficient policies for mitigating motorcycle crash injuries.

The remaining portions of the paper are structured as follows: Section 2 presents reviews on important contributing factors and the used methodological approaches in prior studies of motorcycle-related crash injury severities; Section 3 includes a description of the available crash information; Section 4 describes the construction of a methodological framework; Section 5 interprets the results of transferability between daytime and nighttime accidents and temporal stability test result; Section 6 discusses the model estimation results and compares daytime and nighttime crashes; and Section 7 summarizes the findings and offers some policy-related recommendations based on the model output.

## 2. Literature Reviews

### 2.1. Review of Previous Studies on the General Findings on the Severity of Injuries Sustained from Motorcycle Crashes

In terms of the age of riders, previous research revealed that increases in age were positively correlated with incapacitating and fatal injuries [20–23], and riders older than 50 years of age were likewise more likely to suffer serious and fatal injuries [24,25]. On the other hand, young riders (25 or younger) were favorably related with no injury and mild injury [26,27]. Compared to female riders, studies in the past [26–28] discovered that male riders were more likely to sustain incapacitating and fatal injuries; on the other hand, some recent research discovered that male motorcyclists had a decreased probability of passing away and suffering significant injuries as a result of collisions [24,25,29,30]. Interestingly, gender was also discovered to have a heterogeneous impact on the severity outcomes [31]. In terms of riders' health status, impaired riders and riders under the influence of alcohol were found to be strongly related to significant injuries in crashes [17,22–24,32–35]. Similarly, dangerous speed levels or riding exceeding the posted speed limit was discovered to increase the risk of fatalities and serious injuries [17,18,22,26,30,33,35,36]. Furthermore, negligent riding habits such as weaving through traffic and illegal overtaking were linked to serious and fatal injuries [17,25,34,37–39]. Interestingly, prior research produced inconsistent results regarding the impact of pillions on riders. Certain studies found the presence of a pillion to be associated with a higher chance of serious crashes [31,40,41], while other studies [23,34] reported that it was negatively associated with severe and fatal injuries. In contrast, wearing a helmet greatly decreased the risk of death and serious injuries in collisions involving motorcycles [17,25,34,37–39].

In terms of engine power, earlier studies found that motorbikes with strong engines were more likely to cause serious and fatal injuries in collisions [21,36,42]. However, riders using sports bikes were more likely to be involved in non-incapacitating injuries in collisions [22].

In terms of roadway alignments, prior research found that the injuries of motorcyclists were noticeably worse when collisions occurred on a horizontally curved road [24, 27,34,37,43], slope, or gradient road [17,31,37,41]. Compared to collisions on urban routes, riders involved in collisions on rural roads were more likely to die or suffer serious injuries [26,34,35,39,44]. Regarding intersections, prior research generated contradictory results. Other studies revealed that motorcycle crashes at intersections resulted in a reduced likelihood of death and serious injuries [22,24,37,38], whereas Tamakloe et al. [45] and Vajari et al. [44] reported that motorcycle crashes at junctions increased the risk of mortality and serious injuries. Similarly to this, some studies concluded that motorcycle accidents within U-turn zones were found to increase the risk of more severe injuries [31,41], whereas other studies came to different conclusions [46,47]. Compared to crashes on main lanes or highways lacking frontage lanes, previous research discovered that crashes on auxiliary or frontage lanes had decreased probabilities of fatalities and serious injuries [25,41]. Concerning pavement conditions, prior research interestingly discovered that motorcycle crashes happening in locations with good pavement surface conditions had a higher risk of

fatalities and severe injuries, compared to those occurring in locations with poor pavement conditions [24,25].

Riders engaged in collisions on wet roads were found to have an increased risk of fatality and severe injuries [22,29]; however, motorcycle accidents in poor weather conditions (rain, fog, snow, etc.) were more likely to result in mild or no injuries [23,31]. Regarding the time of day, motorcycle crashes in the daytime were determined to have a decreased risk of fatalities and severe injuries, in comparison with nighttime incidents [20,37,48]. In terms of the day of the week, motorcycle accidents on weekends were reported to be more likely to result in severe and fatal injuries, compared to motorcycle accidents during weekdays [23,27,30].

Prior research revealed that riders who hit fixed objects and skidded off the roads (or rolled over) had a higher chance of fatality and severe injuries [22,23,25,26,36,49]. Additionally, riders who were engaged in collisions with heavy vehicles (such as buses and trucks) had a considerably increased chance of suffering fatal injuries [33,42], and high-impact crash types like angular and head-on crashes were also found to increase the likelihood of fatality [33,36,42].

From the review mentioned above, some accords and disputes can be seen in the existing literature on the severity of motorcycle crashes. This could be a result of four main factors: the use of various statistical methods, variations in crash population sizes, variations in locations where empirical data were obtained, and variations in the times when the data were collected [15,17,50].

*2.2. Review of Motorcycle Crash Severity Modeling Methodologies*

Table 1 provides a list of the methodological approaches used in prior research to evaluate the seriousness of injury in motorcycle crashes. These methods can be divided into three groups: ordered response models, unordered response models, and data-driven methods. The choice of analysis method frequently involves an implicit trade-off between big-data suitability, the predictive capability of the resulting analysis (data-driven methods typically offer this capability better than econometric methods), and inference capability (i.e., the ability to uncover the underlying effect and causality of crash-contributing factors; econometric approaches generally offer this capability better than data-driven approaches) [51]. As presented in Table 1, in recognition of the importance of accounting for unobserved heterogeneity [50], previous motorcycle crash-related studies have used the mixed-ordered logit model [20,37], correlated mixed-ordered probit model with means heterogeneity [41], latent class ordered probit model [34], latent class clustering and latent segmentation based on ordered logit models [18], mixed logit model [30,38], mixed logit/probit model with means and variance heterogeneity [17,36,43,46,52], and latent class multinomial logit model [26]. The selection process between an uncorrelated model (particularly, allowing variance heterogeneity) and a correlated model (allowing interaction among random parameters) should depend on the practical application of the research objective by considering the most appropriate trade-off between flexibility in tracking underlying unobserved effects, predictive power, statistical model fit, and causality or inference capabilities [53,54]. Additionally, picking between the unordered and ordered response econometric methodologies may be difficult considering that both approaches have advantages and drawbacks [55]. However, due to the ordered character of crash injury severity levels [56], the ordered nature of injury severity is accommodated in the statistical modeling procedure.

**Table 1.** Methods utilized in the previous motorcycle crash severity studies.

| Methodological Approach | Previous Research |
| --- | --- |
| *Ordered response models* | |
| Ordered logit model | Albalate and Fernández-Villadangos [32]; Rifaat et al. [33]; Sivasankaran et al. [47] |

**Table 1.** *Cont.*

| Methodological Approach | Previous Research |
|---|---|
| Ordered probit model | Quddus et al. [48]; Chung et al. [40] |
| Generalized ordered logit model | Rifaat et al. [33] |
| Geographically–Temporally Weighted Ordered Logistic Regression | Li et al. [57] |
| Mixed ordered logit model | Chang et al. [37]; Cunto and Ferreira [20] |
| Correlated random parameters ordered probit with heterogeneity in means model | Se et al. [41] |
| Latent class ordered probit model | Li et al. [34] |
| Latent class clustering and latent segmentation based on ordered logit models | Chang et al. [18] |
| *Unordered response models* | |
| Binary logit model | Pai [21]; Rahman et al. [58] |
| Univariate and multivariate stepwise logistic regression model | Zambon and Hasselberg [35] |
| Empirical Bayesian method based on the Multinomial-Dirichlet model | De Lapparent [42] |
| Nested logit model | Savolainen and Mannering [22] |
| Multinomial logit model | Savolainen and Mannering [22]; Schneider and Savolainen [23]; Geedipally et al. [24]; Jung et al. [29]; Wahab and Jiang [59]; Vajari et al. [44] |
| Mixed-effect logit model | Xin et al. [25] |
| Random parameters logit model | Shaheed et al. [30]; Islam and Brown [38] |
| Random parameters binary probit with heterogeneity in means and variance model | Se et al. [31] |
| Random parameters logit with heterogeneity in means and variance model | Waseem et al. [36]; Alnawmasi and Mannering [17]; Ijaz et al. [46]; Islam [43] |
| Latent class multinomial logit model | Shaheed and Gkritza [26] |
| *Data-driven approaches* | |
| Classification and Regression Trees (CART) model | Kashani et al. [39]; Rezapour et al. [60] |
| Artificial Neural Networks (ANN) model | Se et al. [31] |
| Deep learning techniques | Rezapour et al. [61] |

## 3. Empirical Setting

This study makes use of information that was requested from the police accident record department under Thailand's Department of Highways [62] which now maintains jurisdiction over 70% of the nation's roads. This paper cleaned and vetted only data with comprehensive information on motorcycle crash cases. Incomplete data were removed. The used data span from 1 January 2016 until 31 December 2019. As a result, there were 13,795 motorcycle crash cases with complete, detailed information which were then divided into annual daytime and nighttime data. Daytime crash data contained 38 attributes, whereas nighttime data contained 39 attributes (the additional component is "unlit road"). These attributes were categorized into four types of variables, namely, rider characteristics, roadway characteristics, environmental characteristics, and crash characteristics. Three levels of motorcyclist injury severities were considered in this study: minor injury (minor injury or PDO (property damage only (or no injury crash)), severe injury (completely recovered from the injuries incurred after 3 weeks or more), and fatal injury (died at the crash scene or the hospital).

During the daytime, there were 8157 cases (59.1%) of motorcycle crashes, while during the nighttime there were 5638 cases (40.9%). As illustrated in Figure 3, despite the proportions of severe injury crashes in daytime and nighttime being roughly equal (20% relative to their respective total number of crashes), the proportion of fatal injuries resulting from nighttime crashes was noticeably higher than that of daytime crashes (40.21% in nighttime compared to 26.42% in the daytime). This generally showed that drivers involved in nighttime crashes were more likely to die in the collision than drivers involved in daytime incidents. Additionally, the shift in proportion and frequency of each motorcyclist's injury severity from one year to the next was also noted. Table 2 provides a summary of the descriptive statistics of the variables utilized in the model estimation procedure.

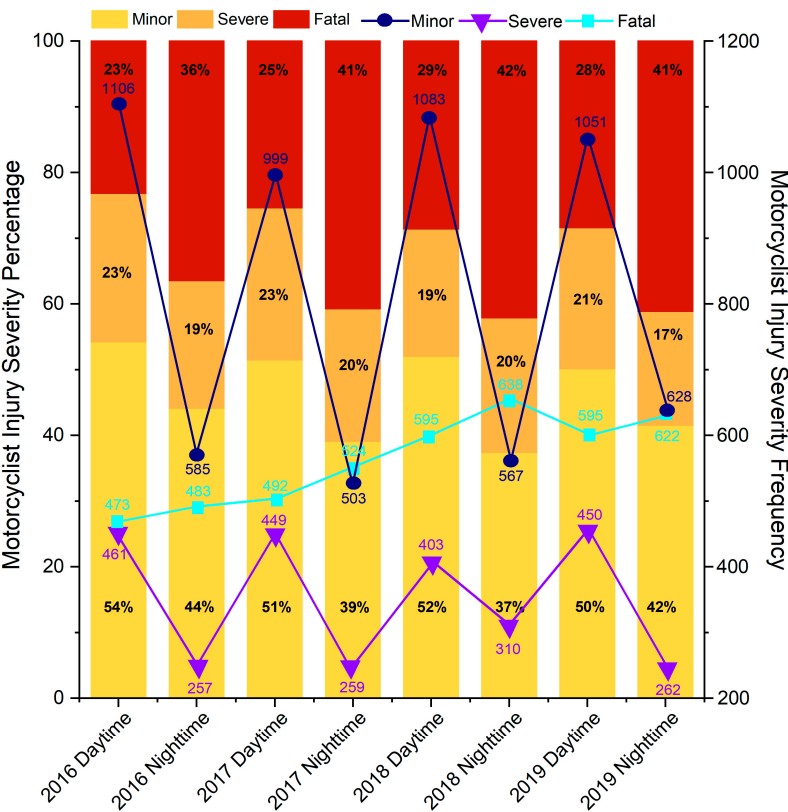

**Figure 3.** Daytime and nighttime motorcyclist injury severity distribution and frequency in Thailand over the years 2016–2019.

**Table 2.** Descriptive statistics of the explanatory variables.

| Explanatory Variable | 2016 | | | | 2017 | | | | 2018 | | | | 2019 | | | |
|---|---|---|---|---|---|---|---|---|---|---|---|---|---|---|---|---|
| | Daytime | | Nighttime | | Daytime | | Nighttime | | Daytime | | Nighttime | | Daytime | | Nighttime | |
| | Mean | SD | Mean | SD | Mean | SD | Mean | SD | Mean | SD | Mean | SD | Mean | SD | Mean | SD |
| *Rider characteristics* | | | | | | | | | | | | | | | | |
| Gender (1 if male, 0 female) | 0.761 | 0.427 | 0.851 | 0.356 | 0.739 | 0.439 | 0.869 | 0.338 | 0.740 | 0.439 | 0.855 | 0.352 | 0.745 | 0.436 | 0.849 | 0.358 |
| Pillion (1 if yes, 0 otherwise) | 0.340 | 0.474 | 0.313 | 0.464 | 0.325 | 0.468 | 0.305 | 0.461 | 0.329 | 0.470 | 0.314 | 0.464 | 0.329 | 0.470 | 0.326 | 0.469 |
| Exceeding speed limit (1 if yes, 0 otherwise) | 0.638 | 0.481 | 0.702 | 0.458 | 0.568 | 0.495 | 0.676 | 0.468 | 0.593 | 0.491 | 0.651 | 0.477 | 0.565 | 0.496 | 0.679 | 0.467 |
| Hit a crossing object (1 if yes, 0 otherwise) | 0.264 | 0.441 | 0.141 | 0.348 | 0.297 | 0.457 | 0.160 | 0.367 | 0.291 | 0.454 | 0.170 | 0.376 | 0.313 | 0.464 | 0.158 | 0.365 |
| Illegal overtaking (1 if yes, 0 otherwise) | 0.017 | 0.128 | 0.011 | 0.102 | 0.016 | 0.127 | 0.014 | 0.118 | 0.013 | 0.113 | 0.013 | 0.111 | 0.010 | 0.100 | 0.007 | 0.081 |
| Alcohol (1 if yes, 0 otherwise) | 0.027 | 0.162 | 0.075 | 0.263 | 0.039 | 0.193 | 0.081 | 0.273 | 0.033 | 0.179 | 0.084 | 0.278 | 0.029 | 0.167 | 0.079 | 0.270 |
| Fatigue (1 if yes, 0 otherwise) | 0.006 | 0.076 | 0.006 | 0.077 | 0.013 | 0.113 | 0.009 | 0.092 | 0.011 | 0.102 | 0.009 | 0.096 | 0.012 | 0.111 | 0.011 | 0.102 |

**Table 2.** *Cont.*

| Explanatory Variable | 2016 | | | | 2017 | | | | 2018 | | | | 2019 | | | |
|---|---|---|---|---|---|---|---|---|---|---|---|---|---|---|---|---|
| | Daytime | | Nighttime | | Daytime | | Nighttime | | Daytime | | Nighttime | | Daytime | | Nighttime | |
| | Mean | SD | Mean | SD | Mean | SD | Mean | SD | Mean | SD | Mean | SD | Mean | SD | Mean | SD |
| *Roadway characteristics* | | | | | | | | | | | | | | | | |
| Main lane (1 if yes, 0 otherwise) | 0.046 | 0.209 | 0.053 | 0.224 | 0.055 | 0.228 | 0.067 | 0.250 | 0.070 | 0.255 | 0.063 | 0.244 | 0.023 | 0.150 | 0.031 | 0.174 |
| Frontage lane (1 if yes, 0 otherwise) | 0.057 | 0.233 | 0.057 | 0.231 | 0.059 | 0.236 | 0.058 | 0.234 | 0.042 | 0.200 | 0.044 | 0.206 | 0.040 | 0.195 | 0.042 | 0.200 |
| Work zone (1 if yes, 0 otherwise) | 0.025 | 0.158 | 0.035 | 0.183 | 0.021 | 0.142 | 0.018 | 0.133 | 0.015 | 0.121 | 0.020 | 0.142 | 0.015 | 0.121 | 0.026 | 0.161 |
| 2 lanes (1 if yes, 0 otherwise) | 0.347 | 0.476 | 0.330 | 0.470 | 0.352 | 0.478 | 0.312 | 0.463 | 0.369 | 0.483 | 0.338 | 0.473 | 0.344 | 0.475 | 0.340 | 0.474 |
| 4 lanes (1 if yes, 0 otherwise) | 0.404 | 0.491 | 0.387 | 0.487 | 0.424 | 0.494 | 0.416 | 0.493 | 0.391 | 0.488 | 0.404 | 0.491 | 0.442 | 0.497 | 0.413 | 0.493 |
| Flush median (1 if yes, 0 otherwise) | 0.082 | 0.274 | 0.057 | 0.231 | 0.105 | 0.307 | 0.096 | 0.295 | 0.112 | 0.316 | 0.119 | 0.324 | 0.135 | 0.342 | 0.115 | 0.319 |
| Raised median (1 if yes, 0 otherwise) | 0.206 | 0.404 | 0.239 | 0.427 | 0.210 | 0.407 | 0.222 | 0.415 | 0.262 | 0.440 | 0.283 | 0.450 | 0.244 | 0.429 | 0.249 | 0.432 |
| Depressed median (1 if yes, 0 otherwise) | 0.160 | 0.367 | 0.162 | 0.368 | 0.164 | 0.371 | 0.174 | 0.379 | 0.174 | 0.379 | 0.171 | 0.377 | 0.198 | 0.399 | 0.208 | 0.406 |
| Barrier median (1 if yes, 0 otherwise) | 0.116 | 0.320 | 0.123 | 0.329 | 0.093 | 0.291 | 0.124 | 0.330 | 0.069 | 0.253 | 0.082 | 0.274 | 0.064 | 0.245 | 0.075 | 0.263 |
| Concrete road (1 if yes, 0 otherwise) | 0.106 | 0.308 | 0.124 | 0.329 | 0.090 | 0.286 | 0.111 | 0.314 | 0.132 | 0.338 | 0.129 | 0.335 | 0.126 | 0.332 | 0.134 | 0.340 |
| Curve (1 if yes, 0 otherwise) | 0.097 | 0.296 | 0.100 | 0.301 | 0.115 | 0.320 | 0.114 | 0.318 | 0.134 | 0.340 | 0.094 | 0.292 | 0.115 | 0.318 | 0.102 | 0.303 |
| Grade (1 if yes, 0 otherwise) | 0.030 | 0.170 | 0.026 | 0.160 | 0.033 | 0.179 | 0.028 | 0.165 | 0.033 | 0.179 | 0.024 | 0.154 | 0.030 | 0.171 | 0.025 | 0.157 |
| 4-leg intersection (1 if yes, 0 otherwise) | 0.046 | 0.209 | 0.041 | 0.198 | 0.046 | 0.210 | 0.030 | 0.172 | 0.047 | 0.211 | 0.040 | 0.195 | 0.050 | 0.218 | 0.035 | 0.184 |
| 3-leg intersection (1 if yes, 0 otherwise) | 0.087 | 0.282 | 0.056 | 0.230 | 0.087 | 0.281 | 0.054 | 0.225 | 0.079 | 0.270 | 0.058 | 0.234 | 0.057 | 0.232 | 0.048 | 0.214 |
| U-turn (1 if yes, 0 otherwise) | 0.077 | 0.267 | 0.072 | 0.258 | 0.101 | 0.301 | 0.075 | 0.263 | 0.085 | 0.279 | 0.055 | 0.229 | 0.065 | 0.247 | 0.035 | 0.184 |
| Bridge (1 if yes, 0 otherwise) | 0.011 | 0.106 | 0.017 | 0.128 | 0.010 | 0.099 | 0.008 | 0.088 | 0.011 | 0.105 | 0.011 | 0.102 | 0.009 | 0.092 | 0.013 | 0.111 |
| Urban road (1 if yes, 0 otherwise) | 0.227 | 0.419 | 0.252 | 0.434 | 0.180 | 0.384 | 0.216 | 0.412 | 0.202 | 0.401 | 0.192 | 0.394 | 0.186 | 0.389 | 0.194 | 0.396 |
| *Environmental characteristics* | | | | | | | | | | | | | | | | |
| Wet road (1 if yes, 0 otherwise) | 0.031 | 0.173 | 0.036 | 0.187 | 0.057 | 0.232 | 0.082 | 0.275 | 0.042 | 0.200 | 0.065 | 0.246 | 0.031 | 0.172 | 0.047 | 0.212 |
| Raining (1 if yes, 0 otherwise) | 0.025 | 0.156 | 0.048 | 0.213 | 0.057 | 0.232 | 0.103 | 0.305 | 0.038 | 0.191 | 0.067 | 0.250 | 0.029 | 0.168 | 0.075 | 0.263 |
| Holiday (1 if yes, 0 otherwise) | 0.446 | 0.497 | 0.455 | 0.498 | 0.443 | 0.497 | 0.482 | 0.500 | 0.454 | 0.498 | 0.510 | 0.500 | 0.406 | 0.491 | 0.462 | 0.499 |
| Weekend (1 if yes, 0 otherwise) | 0.301 | 0.459 | 0.297 | 0.457 | 0.296 | 0.457 | 0.325 | 0.469 | 0.258 | 0.437 | 0.306 | 0.461 | 0.288 | 0.453 | 0.332 | 0.471 |
| Unlit (1 if yes, 0 otherwise) | - | - | 0.285 | 0.451 | - | - | 0.288 | 0.453 | - | - | 0.276 | 0.447 | - | - | 0.282 | 0.450 |
| *Crash characteristics* | | | | | | | | | | | | | | | | |
| Hit a motorcycle (1 if yes, 0 otherwise) | 0.142 | 0.349 | 0.097 | 0.296 | 0.119 | 0.323 | 0.111 | 0.314 | 0.127 | 0.333 | 0.094 | 0.292 | 0.110 | 0.313 | 0.105 | 0.307 |
| Hit a passenger car (1 if yes, 0 otherwise) | 0.309 | 0.462 | 0.257 | 0.437 | 0.273 | 0.445 | 0.236 | 0.425 | 0.292 | 0.455 | 0.217 | 0.412 | 0.260 | 0.438 | 0.215 | 0.411 |
| Hit a pickup truck (1 if yes, 0 otherwise) | 0.282 | 0.450 | 0.207 | 0.405 | 0.299 | 0.458 | 0.196 | 0.397 | 0.307 | 0.461 | 0.217 | 0.412 | 0.350 | 0.477 | 0.247 | 0.431 |
| Hit a van/minibus (1 if yes, 0 otherwise) | 0.051 | 0.220 | 0.052 | 0.222 | 0.057 | 0.231 | 0.049 | 0.216 | 0.048 | 0.213 | 0.041 | 0.198 | 0.031 | 0.173 | 0.029 | 0.168 |
| Hit a truck (1 if yes, 0 otherwise) | 0.080 | 0.272 | 0.103 | 0.304 | 0.087 | 0.281 | 0.117 | 0.321 | 0.086 | 0.280 | 0.109 | 0.312 | 0.092 | 0.289 | 0.106 | 0.309 |
| Rear-end crash (1 if yes, 0 otherwise) | 0.353 | 0.478 | 0.316 | 0.465 | 0.354 | 0.478 | 0.333 | 0.471 | 0.370 | 0.483 | 0.318 | 0.466 | 0.378 | 0.485 | 0.349 | 0.477 |
| Sideswipe crash (1 if yes, 0 otherwise) | 0.247 | 0.431 | 0.177 | 0.381 | 0.239 | 0.427 | 0.170 | 0.376 | 0.298 | 0.458 | 0.207 | 0.405 | 0.280 | 0.449 | 0.200 | 0.400 |
| Single-motorcycle crash (1 if yes, 0 otherwise) | 0.159 | 0.366 | 0.248 | 0.432 | 0.183 | 0.387 | 0.257 | 0.437 | 0.165 | 0.372 | 0.296 | 0.457 | 0.173 | 0.378 | 0.295 | 0.456 |
| Head-on crash (1 if yes, 0 otherwise) | 0.049 | 0.215 | 0.042 | 0.200 | 0.054 | 0.226 | 0.051 | 0.219 | 0.102 | 0.303 | 0.085 | 0.279 | 0.086 | 0.280 | 0.080 | 0.271 |

Note: Using the mean value, readers can extract the distributional percentage (100 × mean) and frequency (total number crashes in each sub-dataset from Figure 3 × mean) of each attribute.

## 4. Methods

To account for unobserved heterogeneity and the ordering nature of motorcycling injury severities, this study used a mixed-ordered probit model that allows potential

variability in means and variances of random parameters. Initially, the model estimation introduces a utility function, $Y_{in}^*$, that determines the likelihood of rider injury severity outcome $i$ in a crash $n$, which is stated as follows [63]:

$$Y_{in}^* = \beta_n X_{in} + \varepsilon_{in} \tag{1}$$

where $\beta_n$ is the vector of estimated parameters, $X_{in}$ signifies the vector of explanatory variables, and $\varepsilon_{in}$ denotes the disturbance term, which is assumed to be normally distributed with a mean of 0 and variance of 1. For crash $n$, the driver's injury severity $Y_n^*$ sustaining injury severity $i$ can be defined as the following [63]:

$$Y_n^* = i, \text{ if } \mu_{i-1,n} < Y_n^* \leq \mu_{i,n} \tag{2}$$

where $i$ ($i$ = 0, 1, or 2, respectively, for minor injury, serious injury, or fatal injury) and $\mu_i$ are estimable parameters (known as thresholds) that define $Y_n^*$, which corresponds to the integer-ordered injury severity level such that $\mu_{i-1} < \mu_i$. The ordered probability $P(y = i)$ of the $i$-th injury severity level for each accident observation is defined as follows [63]:

$$P(y = i) = \Phi(\mu_i - \beta_n X_n) - \Phi(\mu_{i+1} - \beta_n X_n) \tag{3}$$

where $\Phi(.)$ denotes the cumulative standard normal distribution. To allow explanatory variables to influence means and variances of the random parameter, let $\beta_{in}$ be a vector of estimable parameters that vary across crashes as follows [64]:

$$\beta_{in} = \beta_i + \psi_{in}\Omega_{in} + \sigma_{in}e^{(\gamma_{in}\Psi_{in})}\omega_{in}, \tag{4}$$

where $\beta_i$ is the mean value of the random parameter vector, $\Omega_{in}$ is a vector of attributes that captures heterogeneity in the mean, $\psi_{in}$ is the corresponding vector of estimable parameters, $\Psi_{in}$ is a vector of attributes that captures heterogeneity in the standard deviation $\sigma_{in}$ with the corresponding parameter vector $\gamma_{in}$, and $\omega_{in}$ are vectors of randomly distributed terms. The Halton sequence procedure is employed for a simulated maximum likelihood estimation process to make parameter estimation computationally effective and trustworthy [65]. To achieve this, the paper estimated the models using maximum likelihood estimation with 1000 Halton draws [66,67]. Much as it had been in earlier studies [9,17,68], normal distribution was taken into consideration for the function form of the parameter density function, as it often provides the greatest fit for data on injury severities. To make the interpretation of the results easier, the current study also aims to compute the marginal effects to analyze the effect of the explanatory variables on the probability of each injury severity level where the direction of the effects cannot be represented by the parameter estimates [55]. Marginal effects are calculated by the change in the resulting probability of each ordered outcome due to a one-unit change in the explanatory variable (i.e., change from "0" to "1" in the case of indicator variables). In this investigation, marginal effects are calculated by averaging over observations as follows [55,63]:

$$\frac{P(y = i)}{\partial X} = [\Phi(\mu_{i-1} - \beta X) - \Phi(\mu_i - \beta X)]\beta \tag{5}$$

## 5. Temporal Stability and Transferability Test

In this section, the study conducted two rounds of likelihood ratio tests to determine the level of significant difference between sub-models. The tests are run to evaluate whether the following null hypotheses should be accepted or rejected:

**H1.** *The impacts of parameter estimates are the same in the daytime and nighttime motorcycle crashes.*

**H2.** *The impacts of parameter estimates are temporally stable from one year to the next.*

These tests can be calculated using the following equation [63]:

$$X^2 = -2\big[LL\big(\beta_{k_2 k_1}\big) - LL\big(\beta_{k_1}\big)\big] \tag{6}$$

where $k_2$ and $k_1$ stand for two distinct sub-models employing distinct sub-datasets. $LL\big(\beta_{k_2 k_1}\big)$ denotes the log-likelihood function of a model that incorporates significant indicators from $k_2$ while using data from the subset $k_1$, and $LL\big(\beta_{k_1}\big)$ denotes the log-likelihood function of the original model $k_1$. These likelihood tests are also flipped, so that $k_2$ becomes $k_1$ and $k_1$ becomes $k_2$. To establish the significance level or confidence level, a degree of freedom equal to the number of estimated parameters is used. The outcomes of the tests for transferability (between daytime and nighttime) and temporal stability are demonstrated in Tables 3 and 4, respectively. The results in Table 3 reject H1 with over 95% for each year. In addition, Table 4 demonstrates that most of the 2-year pair-wise tests also reject H2 with a relatively high degree of confidence (19 out of 24 tests yield confidence levels of more than 99%). Given that both assumptions were disproved, daytime and nighttime crashes should be modeled separately, and yearly instability should also be taken into consideration.

**Table 3.** Likelihood ratio test results between daytime and nighttime motorcyclist injury severity models for different years (chi-square, degree of freedom in bracket, and confidence level in parenthesis).

| Years | T1= | Daytime | Nighttime |
|---|---|---|---|
| | T2= | Nighttime | Daytime |
| 2016 | | 134.72 [17] (99.99%) | 100.31 [21] (99.99%) |
| 2017 | | 31.93 [18] (97.76%) | 61.17 [20] (99.99%) |
| 2018 | | 79.21 [23] (99.99%) | 40.76 [25] (97.57%) |
| 2019 | | 62.72 [26] (99.99%) | 112.81 [19] (99.99%) |

**Table 4.** Temporal stability test results of daytime and nighttime motorcyclist injury severity models (chi-square, degree of freedom in bracket, and confidence level in parenthesis).

| Y1/Y2 | 2016 | | 2017 | | 2018 | | 2019 | |
|---|---|---|---|---|---|---|---|---|
| | Daytime | Nighttime | Daytime | Nighttime | Daytime | Nighttime | Daytime | Nighttime |
| 2016 | - | - | 89.53 [20] (99.99%) | 113.55 [19] (99.99%) | 30.25 [25] (78.52%) | 98.73 [24] (99.99%) | 102.91 [19] (99.99%) | 92.73 [27] (99.99%) |
| 2017 | 64.22 [21] (99.99%) | 90.78 [18] (99.99%) | - | - | 11.28 [25] (0.85%) | 55.48 [24] (99.93%) | 78.21 [19] (99.99%) | 70.31 [27] (99.99%) |
| 2018 | 74.09 [21] (99.99%) | 64.96 [18] (99.99%) | 70.51 [20] (99.99%) | 71.58 [19] (99.99%) | - | - | 117.83 [19] (99.99%) | 55.62 [27] (99.90%) |
| 2019 | 42.71 [21] (99.49%) | 116.61 [18] (99.99%) | 54.34 [20] (99.99%) | 105.71 [19] (99.99%) | 28.38 [25] (70.96%) | 77.72 [24] (99.99%) | - | - |

## 6. Results and Discussion

Tables 5 and 6 show the 2016-through-2019 estimation results of the mixed-ordered probit model with means and variance heterogeneity for daytime and nighttime, respectively. To better highlight the distinction and facilitate the comprehension of the model results, the descriptions of the marginal effect of the significant variables are presented in Tables 7 and 8 for the daytime and nighttime models, respectively. It should be noted that, for random parameters with significant standard deviations and insignificant means, likelihood ratio tests ($X^2 = -2[LL(\beta_{model\ with\ RP}) - LL(\beta_{model\ without\ RP})]$ distributed with 2 degrees of freedom) were carried out to see if the improvement of the model fit was statistically significant. In this study, random parameters were retained only if they showed improvement in the model fit at 0.1 or lower.

**Table 5.** Estimation results of mixed-ordered probit models with heterogeneity in the means and variances for daytime motorcycle crash severity models (bolds indicate random parameter).

| Variables | 2016 | | 2017 | | 2018 | | 2019 | |
|---|---|---|---|---|---|---|---|---|
| | Coef. | t-Stat | Coef. | t-Stat | Coef. | t-Stat | Coef. | t-Stat |
| *Rider characteristics* | | | | | | | | |
| Pillion | 0.161 | 2.53 | **0.793** | **6.06** | 0.275 | 3.76 | **0.242** | **3.43** |
| SD "Pillion" | | | **0.533** | **9.47** | | | **0.825** | **9.97** |
| Exceeding speed limit | | | −0.253 | −1.99 | −0.028 | −0.13 | | |
| SD "Exceeding speed limit" | | | **0.936** | **17.99** | **1.301** | **21.87** | | |
| Illegal overtaking | 0.652 | 3.01 | 0.657 | 2.81 | 1.205 | 4.26 | | |
| *Roadway characteristics* | | | | | | | | |
| Frontage lane | −0.448 | −2.46 | −1.554 | −6.62 | −0.969 | −4.55 | | |
| Work zone | | | −0.406 | −1.90 | | | **−0.339** | **−1.20** |
| SD "Work zone" | | | | | | | **1.339** | **2.78** |
| 2 lanes | 0.649 | 5.07 | | | | | 2.210 | 4.92 |
| 4 lanes | 0.413 | 4.18 | | | | | 0.382 | 4.09 |
| Flush median | | | 0.873 | 2.09 | 0.742 | 2.34 | | |
| Raised median | **−0.189** | **−1.16** | | | | | −0.171 | −2.11 |
| SD "Raised median" | **1.377** | **13.68** | | | | | | |
| Depressed median | | | **0.748** | **1.70** | 0.590 | 3.76 | | |
| SD "Depressed median" | | | **0.497** | **6.19** | | | | |
| Concrete road | | | −0.258 | −2.18 | −0.309 | −2.76 | | |
| Curve | 0.260 | 2.49 | | | | | | |
| Grade | | | | | 0.522 | 2.89 | 0.350 | 1.83 |
| 4-leg intersection | | | | | 0.755 | 4.76 | | |
| 3-leg intersection | | | | | | | −0.219 | −1.81 |
| U-turn | 0.373 | 3.43 | | | | | 0.341 | 2.95 |
| Urban road | −0.277 | −3.38 | **−0.751** | **−3.73** | −0.473 | −5.17 | −0.133 | −1.66 |
| SD "Urban road" | | | **1.283** | **11.15** | | | | |
| *Environmental characteristics* | | | | | | | | |
| Raining | | | | | 1.670 | 2.64 | | |
| Weekend | 0.257 | 2.74 | 0.438 | 4.52 | 0.254 | 2.64 | 0.382 | 3.83 |
| *Crash characteristics* | | | | | | | | |
| Hit a motorcycle | −0.490 | −4.67 | −0.724 | −5.93 | −0.467 | −3.57 | −0.525 | −4.62 |
| Hit a pickup truck | 0.220 | 2.56 | **−0.275** | **−1.78** | 0.339 | 3.20 | | |
| SD "Hit a pickup truck" | | | **1.003** | **14.22** | | | | |
| Hit a van/minibus | 0.499 | 3.53 | 0.362 | 2.53 | 0.663 | 3.84 | | |
| Hit a truck | **0.818** | **3.94** | 1.204 | 9.38 | 1.701 | 11.65 | 0.846 | 7.36 |
| SD "Hit a truck" | **1.248** | **8.61** | | | | | | |
| Rear-end crash | −0.287 | −3.41 | | | −0.755 | −5.31 | | |
| Sideswipe crash | **−0.336** | **−2.77** | −0.180 | −1.81 | **−1.689** | **−8.10** | −0.308 | −2.71 |
| SD "Sideswipe crash" | **0.750** | **10.80** | | | **1.048** | **13.92** | | |
| Single-motorcycle crash | −0.271 | −2.62 | | | −0.500 | −3.05 | **−0.505** | **−3.68** |
| SD "Single-motorcycle crash" | | | | | | | **0.742** | **4.42** |
| Head-on crash | | | 0.391 | 2.63 | **−0.439** | **−1.06** | 0.266 | 1.87 |
| SD "Head-on crash" | | | | | **1.029** | **9.00** | | |
| *Heterogeneity in mean* | | | | | | | | |
| Raised median: Exceeding speed limit | −0.496 | −3.08 | | | | | | |
| Hit a truck: Flush median | 2.486 | 3.16 | | | | | | |
| Depressed median: Alcohol | | | 0.639 | 3.52 | | | | |
| Exceeding speed limit: 2 lanes | | | | | 0.422 | 1.67 | | |
| Sideswipe crash: 2 lanes | | | | | 0.509 | 2.45 | | |
| Sideswipe crash: 4 lanes | | | | | 0.626 | 3.05 | | |
| Work zone: Depressed median | | | | | | | 2.332 | 2.36 |
| *Heterogeneity in variance* | | | | | | | | |
| Exceeding speed limit: Main lane | | | −2.884 | −13.81 | | | | |
| Hit a pickup truck: Main lane | | | 3.812 | 11.88 | | | | |
| Urban road: Main lane | | | −2.414 | −10.47 | | | | |
| Exceeding speed limit: Wet road | | | | | 0.530 | 3.22 | | |
| Sideswipe crash: Wet road | | | | | 13.208 | 44.49 | | |
| Work zone: Exceeding speed limit | | | | | | | 0.870 | 1.80 |
| Single-motorcycle crash: Exceeding speed limit | | | | | | | 1.512 | 6.08 |
| *Threshold* | | | | | | | | |
| μ | 0.844 | 23.64 | 0.990 | 23.71 | 0.990 | 22.41 | 0.730 | 23.33 |

**Table 5.** *Cont.*

| Variables | 2016 | | 2017 | | 2018 | | 2019 | |
|---|---|---|---|---|---|---|---|---|
| | **Coef.** | **t-Stat** | **Coef.** | **t-Stat** | **Coef.** | **t-Stat** | **Coef.** | **t-Stat** |
| *Model fit statistic* | | | | | | | | |
| Number of observations | 2040 | | 1940 | | 2081 | | 2096 | |
| Log-likelihood at zero | −2054.083 | | −1995.107 | | −2113.879 | | −2167.074 | |
| Log-likelihood at convergence | −1853.152 | | −1818.181 | | −1919.764 | | −1999.817 | |
| McFadden | 0.0978 | | 0.0887 | | 0.0918 | | 0.0772 | |

**Table 6.** Estimation results of mixed-ordered probit models with heterogeneity in the means and variances for nighttime motorcycle crash severity models (bolds indicate random parameter).

| Variables | 2016 | | 2017 | | 2018 | | 2019 | |
|---|---|---|---|---|---|---|---|---|
| | **Coef.** | **t-Stat** | **Coef.** | **t-Stat** | **Coef.** | **t-Stat** | **Coef.** | **t-Stat** |
| *Rider characteristics* | | | | | | | | |
| Gender | −0.355 | −2.47 | | | 0.457 | 5.10 | 0.417 | 4.16 |
| Pillion | 0.281 | 3.24 | 0.198 | 2.27 | 0.112 | 1.66 | 0.368 | 4.48 |
| Exceeding speed limit | | | | | | | **0.396** | **3.56** |
| SD "Exceeding speed limit" | | | | | | | **0.985** | **17.70** |
| Fatigue | 0.848 | 2.21 | 1.423 | 2.58 | | | 0.739 | 2.09 |
| *Roadway characteristics* | | | | | | | | |
| Frontage lane | −0.400 | −1.87 | **−1.780** | **−5.22** | −1.088 | −5.57 | | |
| SD "Frontage lane" | | | **1.334** | **3.10** | | | | |
| Work zone | | | | | | | 0.587 | 2.65 |
| 2 Lanes | 0.674 | 3.92 | | | 1.575 | 3.13 | 2.593 | 4.61 |
| 4 Lanes | **−0.850** | **−3.95** | | | 0.179 | 1.73 | 0.365 | 3.57 |
| SD "4 Lanes" | **0.597** | **8.18** | | | | | | |
| Flush median | | | 0.443 | 3.15 | 0.318 | 2.63 | | |
| Raised median | | | | | 0.279 | 2.91 | | |
| Depressed median | | | **0.249** | **1.71** | | | | |
| SD "Depressed median" | | | **0.770** | **4.51** | | | | |
| Concrete road | 0.288 | 2.47 | | | −0.274 | −2.32 | | |
| SD "Concrete road" | | | | | **0.749** | **6.26** | | |
| Grade | 0.564 | 2.42 | 0.444 | 1.73 | | | | |
| 3-leg intersection | | | | | 0.264 | 1.83 | | |
| U-turn | 0.461 | 3.28 | | | | | | |
| Urban road | | | −0.159 | −1.67 | −0.199 | −2.16 | **−0.242** | **−1.96** |
| SD "Urban road" | | | | | | | **0.782** | **7.33** |
| *Environmental characteristics* | | | | | | | | |
| Wet road | 0.829 | 2.93 | 0.808 | 2.72 | 1.261 | 4.54 | 0.631 | 2.40 |
| Raining | | | | | −0.787 | −2.75 | **−0.294** | **−1.25** |
| SD "Raining" | | | | | | | **1.142** | **6.13** |
| Weekend | | | 0.606 | 5.56 | 0.235 | 2.49 | | |
| Unlit | 0.357 | 3.91 | 0.174 | 2.02 | 0.209 | 2.57 | 0.274 | 3.22 |
| *Crash characteristics* | | | | | | | | |
| Hit a motorcycle | −0.391 | −2.64 | −0.381 | −2.83 | −0.387 | −2.79 | −0.815 | −5.32 |
| Hit a passenger car | | | −0.216 | −2.21 | −0.251 | −2.21 | −0.239 | −2.08 |
| Hit a pickup truck | **0.056** | **0.21** | **0.020** | **0.14** | **0.372** | **3.09** | | |
| SD "Hit a pickup truck" | **0.722** | **6.39** | **1.058** | **6.44** | **0.620** | **7.56** | | |
| Hit a van/minibus | | | | | **0.258** | **1.21** | 0.464 | 2.01 |
| SD "Hit a van/minibus" | | | | | **1.358** | **5.13** | | |
| Hit a truck | 0.945 | 6.70 | 0.780 | 5.70 | 0.981 | 6.81 | **1.352** | **7.91** |
| SD "Hit a truck" | | | | | | | **0.343** | **2.63** |
| Rear-end crash | | | | | | | −0.447 | −3.05 |
| Sideswipe crash | −0.528 | −4.18 | | | **−0.274** | **−1.92** | −0.494 | −3.22 |
| SD "Sideswipe crash" | | | | | **0.693** | **7.75** | | |
| Single-motorcycle crash | | | | | | | −0.522 | −3.35 |
| Head-on crash | 0.525 | 2.64 | 0.688 | 3.80 | 0.380 | 2.22 | 0.456 | 2.44 |
| *Heterogeneity in mean* | | | | | | | | |
| 4 lanes: Hit a crossing object | 0.768 | 3.53 | | | | | | |
| Hit a pickup truck: Hit a crossing object | 0.400 | 1.65 | | | | | | |
| Frontage lane: Alcohol | | | 3.806 | 3.75 | | | | |

**Table 6.** *Cont.*

| Variables | 2016 | | 2017 | | 2018 | | 2019 | |
|---|---|---|---|---|---|---|---|---|
| | Coef. | t-Stat | Coef. | t-Stat | Coef. | t-Stat | Coef. | t-Stat |
| Frontage lane: Raised median | | | 2.221 | 2.71 | | | | |
| Depressed median: Alcohol | | | −0.676 | −1.87 | | | | |
| Hit a pickup truck: Curve | | | | | 2.344 | 3.50 | | |
| Sideswipe crash: Curve | | | | | −1.511 | −2.51 | | |
| Urban road: Barrier median | | | | | | | −0.864 | −3.33 |
| Raining: Concrete road | | | | | | | −1.192 | −2.44 |
| Raining: Curve | | | | | | | −1.308 | −2.41 |
| Raining: U-turn | | | | | | | 1.449 | 2.07 |
| Hit a truck: Barrier median | | | | | | | −0.845 | −2.01 |
| Hit a truck: Concrete road | | | | | | | −0.831 | −2.06 |
| Hit a truck: U-turn | | | | | | | −2.137 | −1.84 |
| *Heterogeneity in variance* | | | | | | | | |
| 4 lanes: Rear-end crash | 0.923 | 5.73 | | | | | | |
| Depressed median: Exceeding speed limit | | | −1.363 | −5.21 | | | | |
| Hit a pickup truck: Exceeding speed limit | | | 3.315 | 16.98 | | | | |
| Concrete road: Depressed median | | | | | 1.783 | 4.88 | | |
| Hit a pickup truck: Depressed median | | | | | −1.627 | −4.32 | | |
| Sideswipe crash: Depressed median | | | | | 2.392 | 7.76 | | |
| Exceeding speed limit: Frontage lane | | | | | | | −0.927 | −1.87 |
| Raining: Alcohol | | | | | | | 1.021 | 1.84 |
| Urban road: Alcohol | | | | | | | −2.597 | −6.57 |
| *Threshold* | | | | | | | | |
| μ | 0.705 | 17.67 | 0.670 | 17.57 | 0.681 | 19.34 | 0.677 | 17.71 |
| *Model fit statistic* | | | | | | | | |
| Number of observations | 1325 | | 1286 | | 1515 | | 1512 | |
| Log-likelihood at zero | −1387.194 | | −1357.652 | | −1600.863 | | −1563.527 | |
| Log-likelihood at convergence | −1202.744 | | −1228.335 | | −1429.287 | | −1373.630 | |
| McFadden | 0.1330 | | 0.0953 | | 0.1072 | | 0.1215 | |

**Table 7.** Summary of the marginal effects of significant variables in daytime motorcycle crash severity models (bold values indicate random parameter).

| | 2016 | | | 2017 | | | 2018 | | | 2019 | | |
|---|---|---|---|---|---|---|---|---|---|---|---|---|
| | Minor | Severe | Fatal | Minor | Severe | Fatal | Minor | Severe | Fatal | Minor | Severe | Fatal |
| *Rider characteristics* | | | | | | | | | | | | |
| Pillion | −0.0546 | 0.0137 | 0.0408 | **−0.2499** | **0.0644** | **0.1855** | −0.0784 | 0.0291 | 0.0493 | **−0.0853** | **0.0147** | **0.0705** |
| Exceeding speed limit | | | | **0.0782** | **−0.0243** | **−0.0538** | **0.0079** | **−0.0030** | **−0.0048** | | | |
| Illegal overtaking | −0.2150 | 0.0217 | 0.1933 | −0.1979 | 0.0313 | 0.1666 | −0.3390 | 0.0483 | 0.2906 | | | |
| *Roadway characteristics* | | | | | | | | | | | | |
| Frontage lane | 0.1489 | −0.0535 | −0.0954 | 0.3737 | −0.1988 | −0.1749 | 0.2318 | −0.1168 | −0.1149 | | | |
| Work zone | | | | 0.1214 | −0.0473 | −0.0740 | | | | **0.1176** | **−0.0310** | **−0.0866** |
| 2 Lanes | −0.2173 | 0.0486 | 0.1687 | | | | | | | −0.4772 | −0.0287 | 0.5059 |
| 4 Lanes | −0.1405 | 0.0357 | 0.1048 | | | | | | | −0.1350 | 0.0251 | 0.1099 |
| Flush median | | | | −0.2522 | 0.0327 | 0.2195 | −0.2177 | 0.0623 | 0.1554 | | | |
| Raised median | **0.0641** | **−0.0186** | **−0.0455** | | | | | | | 0.0601 | −0.0127 | −0.0474 |
| Depressed median | | | | **−0.2189** | **0.0382** | **0.1807** | −0.1714 | 0.0549 | 0.1164 | | | |
| Concrete road | | | | 0.0786 | −0.0283 | −0.0503 | 0.0847 | −0.0362 | −0.0484 | | | |
| Curve | −0.0889 | 0.0192 | 0.0696 | | | | | | | | | |
| Grade | | | | | | | −0.1521 | 0.0456 | 0.1064 | −0.1218 | 0.0133 | 0.1085 |
| 4-leg intersection | | | | | | | −0.2203 | 0.0554 | 0.1649 | | | |
| 3-leg intersection | | | | | | | | | | 0.0767 | −0.0180 | −0.0586 |
| U-turn | −0.1266 | 0.0239 | 0.1027 | | | | | | | −0.1189 | 0.0142 | 0.1047 |
| Urban road | 0.0948 | −0.0295 | −0.0652 | **0.2246** | **−0.0967** | **−0.1279** | 0.1289 | −0.0570 | −0.0718 | 0.0468 | −0.0100 | −0.0368 |
| *Environmental characteristics* | | | | | | | | | | | | |
| Raining | | | | | | | −0.4487 | 0.0224 | 0.4262 | | | |
| Weekend | −0.0865 | 0.0205 | 0.0659 | −0.1336 | 0.0368 | 0.0968 | −0.0720 | 0.0262 | 0.0457 | −0.1326 | 0.0199 | 0.1127 |
| *Crash characteristics* | | | | | | | | | | | | |
| Hit a motorcycle | 0.1644 | −0.0569 | −0.1074 | 0.2108 | −0.0859 | −0.1249 | 0.1253 | −0.0554 | −0.0699 | 0.1810 | −0.0502 | −0.1308 |
| Hit a pickup truck | −0.0753 | 0.0187 | 0.0566 | **0.0843** | **−0.0283** | **−0.0560** | −0.0966 | 0.0357 | 0.0609 | | | |
| Hit a van/minibus | −0.1676 | 0.0256 | 0.1419 | −0.1114 | 0.0268 | 0.0845 | −0.1924 | 0.0521 | 0.1402 | | | |
| Hit a truck | **−0.2702** | **0.0219** | **0.2482** | −0.3493 | 0.0128 | 0.3364 | −0.4686 | 0.0285 | 0.4400 | −0.2840 | −0.0010 | 0.2851 |

**Table 7.** *Cont.*

|  | 2016 | | | 2017 | | | 2018 | | | 2019 | | |
| --- | --- | --- | --- | --- | --- | --- | --- | --- | --- | --- | --- | --- |
|  | Minor | Severe | Fatal | Minor | Severe | Fatal | Minor | Severe | Fatal | Minor | Severe | Fatal |
| Rear-end crash | 0.0966 | −0.0273 | −0.0692 |  |  |  | 0.1954 | −0.0717 | −0.1236 |  |  |  |
| Sideswipe crash | **0.1133** | **−0.0346** | **−0.0787** | 0.0552 | −0.0185 | −0.0366 | **0.4045** | **−0.1846** | **−0.2198** | 0.1082 | −0.0241 | −0.0841 |
| Single-motorcycle crash | 0.0906 | −0.0272 | −0.0634 |  |  |  | 0.1338 | −0.0558 | −0.0780 | **0.1759** | −0.0457 | −0.1302 |
| Head-on crash |  |  |  | −0.12047 | 0.0283 | 0.0921 | **0.1153** | **−0.0488** | **−0.0665** | −0.0936 | 0.0129 | 0.0806 |

**Table 8.** Summary of the marginal effects of significant variables in nighttime motorcycle crash severity models (bold values indicate random parameter).

|  | 2016 | | | 2017 | | | 2018 | | | 2019 | | |
| --- | --- | --- | --- | --- | --- | --- | --- | --- | --- | --- | --- | --- |
|  | Minor | Severe | Fatal | Minor | Severe | Fatal | Minor | Severe | Fatal | Minor | Severe | Fatal |
| *Rider characteristics* | | | | | | | | | | | | |
| Gender | 0.1047 | −0.0187 | −0.0860 |  |  |  | −0.1490 | 0.0071 | 0.1419 | −0.1308 | 0.0100 | 0.1208 |
| Pillion | −0.0823 | 0.0166 | 0.0656 | −0.0626 | −0.0020 | 0.0646 | −0.0349 | −0.0013 | 0.0363 | −0.1101 | −0.0022 | 0.1124 |
| Exceeding speed limit |  |  |  |  |  |  |  |  |  | **−0.1231** | **0.0052** | **0.1178** |
| Fatigue | −0.2463 | 0.0189 | 0.2273 | −0.3140 | −0.1142 | 0.4282 |  |  |  | −0.1978 | −0.0318 | 0.2297 |
| *Roadway characteristics* | | | | | | | | | | | | |
| Frontage lane | 0.1124 | −0.0304 | −0.0819 | **0.5234** | **−0.1545** | **−0.3689** | 0.3544 | −0.0664 | −0.2879 |  |  |  |
| Work zone |  |  |  |  |  |  |  |  |  | −0.1639 | −0.0200 | 0.1840 |
| 2 Lanes | −0.2120 | 0.0446 | 0.1674 |  |  |  | −0.3823 | −0.0433 | 0.4256 | −0.4319 | −0.0587 | 0.4907 |
| 4 Lanes | **0.2539** | **−0.0691** | **−0.1848** |  |  |  | −0.0563 | −0.0018 | 0.0582 | −0.1118 | 0.0002 | 0.1116 |
| Flush median |  |  |  | −0.1331 | −0.0137 | 0.1468 | −0.0990 | −0.0028 | 0.1018 |  |  |  |
| Raised median |  |  |  |  |  |  | −0.0892 | 0.0000 | 0.0891 |  |  |  |
| Depressed median |  |  |  | **−0.0775** | **−0.0041** | **0.0817** |  |  |  |  |  |  |
| Concrete road | −0.0840 | 0.0154 | 0.0685 |  |  |  | **0.0889** | **−0.0021** | **−0.0868** |  |  |  |
| Grade | −0.1667 | 0.0212 | 0.1454 | −0.1315 | −0.0163 | 0.1478 |  |  |  |  |  |  |
| 3-leg intersection |  |  |  |  |  |  | −0.0799 | −0.0068 | 0.0867 |  |  |  |
| U-turn | −0.1350 | 0.0206 | 0.1144 |  |  |  |  |  |  |  |  |  |
| Urban road |  |  |  | 0.0515 | −0.0006 | −0.0508 | 0.0638 | −0.0003 | −0.0635 | **0.0754** | **−0.0036** | **−0.0718** |
| *Environmental characteristics* | | | | | | | | | | | | |
| Wet road | −0.2428 | 0.0217 | 0.2210 | −0.2243 | −0.0418 | 0.2662 | −0.2988 | −0.0934 | 0.3923 | −0.1753 | −0.0219 | 0.1973 |
| Raining |  |  |  |  |  |  | 0.2511 | −0.0330 | −0.2180 | **0.0911** | **−0.0060** | **−0.0851** |
| Weekend |  |  |  | −0.1839 | −0.0092 | 0.1931 | −0.0727 | −0.0035 | 0.0762 |  |  |  |
| Unlit | −0.1073 | 0.0215 | 0.0858 | −0.0552 | −0.0016 | 0.0568 | −0.0650 | −0.0033 | 0.0684 | −0.0831 | −0.0013 | 0.0844 |
| *Crash characteristics* | | | | | | | | | | | | |
| Hit a motorcycle | 0.1097 | −0.02843 | −0.08136 | 0.1250 | −0.0072 | −0.1177 | 0.1263 | −0.0057 | −0.1205 | 0.2565 | −0.03570 | −0.22080 |
| Hit a passenger car |  |  |  | 0.0703 | −0.0011 | −0.0692 | 0.0810 | −0.0005 | −0.0804 | 0.0741 | −0.00299 | −0.07120 |
| Hit a pickup truck | **−0.0163** | **0.0034** | **0.0129** | **−0.0063** | **−0.0001** | **0.0064** | **−0.1140** | **−0.0090** | **0.1231** |  |  |  |
| Hit a van/minibus |  |  |  |  |  |  | −0.0778 | −0.0068 | 0.0846 | −0.1329 | −0.0121 | 0.1450 |
| Hit a truck | −0.2821 | 0.0249 | 0.2572 | −0.2223 | −0.0423 | 0.2646 | −0.2601 | −0.0678 | 0.3279 | **−0.3240** | **−0.1001** | **0.4242** |
| Rear-end crash |  |  |  |  |  |  |  |  |  | 0.1349 | −0.0059 | −0.1289 |
| Sideswipe crash | 0.1497 | −0.0417 | −0.1079 |  |  |  | 0.0883 | −0.0011 | −0.0872 | 0.1541 | −0.0115 | −0.1426 |
| Single-motorcycle crash |  |  |  |  |  |  |  |  |  | 0.1595 | −0.0051 | −0.1544 |
| Head-on crash | −0.1554 | 0.0214 | 0.1340 | −0.1941 | −0.0346 | 0.2288 | −0.1136 | −0.0129 | 0.1266 | −0.1325 | −0.0114 | 0.1439 |

It is important to note that the data used for this study were provided by police officers and consisted of historical and secondary data. However, the study did not include information on whether or not the motorcycle riders were wearing helmets, which is an important factor for motorcycle users. This information was not available in the police reports. Despite this limitation, the study identified other significant factors that determined the severity of the injuries sustained in motorcycle crashes, as demonstrated by their significance levels. These significant factors were discussed and compared with the previous literature, and the findings were consistent with previous studies, which adds credibility to the research. It is worth noting that, similar to this study, many previous studies on motorcycle crash injury severity also did not have information on helmet use status due to the unavailability of these data, including the studies by Rifaat et al. [33]; Waseem et al. [36]; Quddus et al. [48]; Pai and Saleh [69]; Pai and Saleh [28]; [21]; Se et al. [52]; Islam [70]; Ding et al. [71]; Li et al. [34]; and Rezapour et al. [60].

## 6.1. Rider Characteristics

With regard to gender, male cyclists were not shown to be statistically significant in all-year daytime models; however, they were significant in 2016, 2018, and 2019 nighttime models. When compared to female riders, the marginal effects show that male riders had a reduced risk of death and serious injuries in 2016, but a higher risk of death and serious injuries in 2018 and 2019 during nighttime. The observed temporal instability in the latest period may be attributed to the changes in male riding attitudes over time toward aggressiveness, alcohol misuse, and unsafe riding behavior [72]. One potential explanation for the lack of significance in the 2017 model could be that the risk-taking behavior of male riders had only begun to increase and have an impact, resulting in no statistically significant difference between male and female riders during that period.

Compared to lone riders, the marginal effect finding reveals that motorcyclists riding with a stable pillion passenger had a higher probability of fatal injuries across all years of both daytime and nighttime models. This indicator also generated random parameters in the 2017 and 2019 daytime models, with 93.16% and 61.59% of the riding-with-pillion crashes expected to result in fatal injuries (Figure 4a,b), respectively. The primary causes may be due to the additional weight of the pillion, which could affect the braking distance and heighten the crash impact [17], and peer pressure of the pillion passenger on the rider's risk-taking behaviors such as excessive speed or aggressive riding [73].

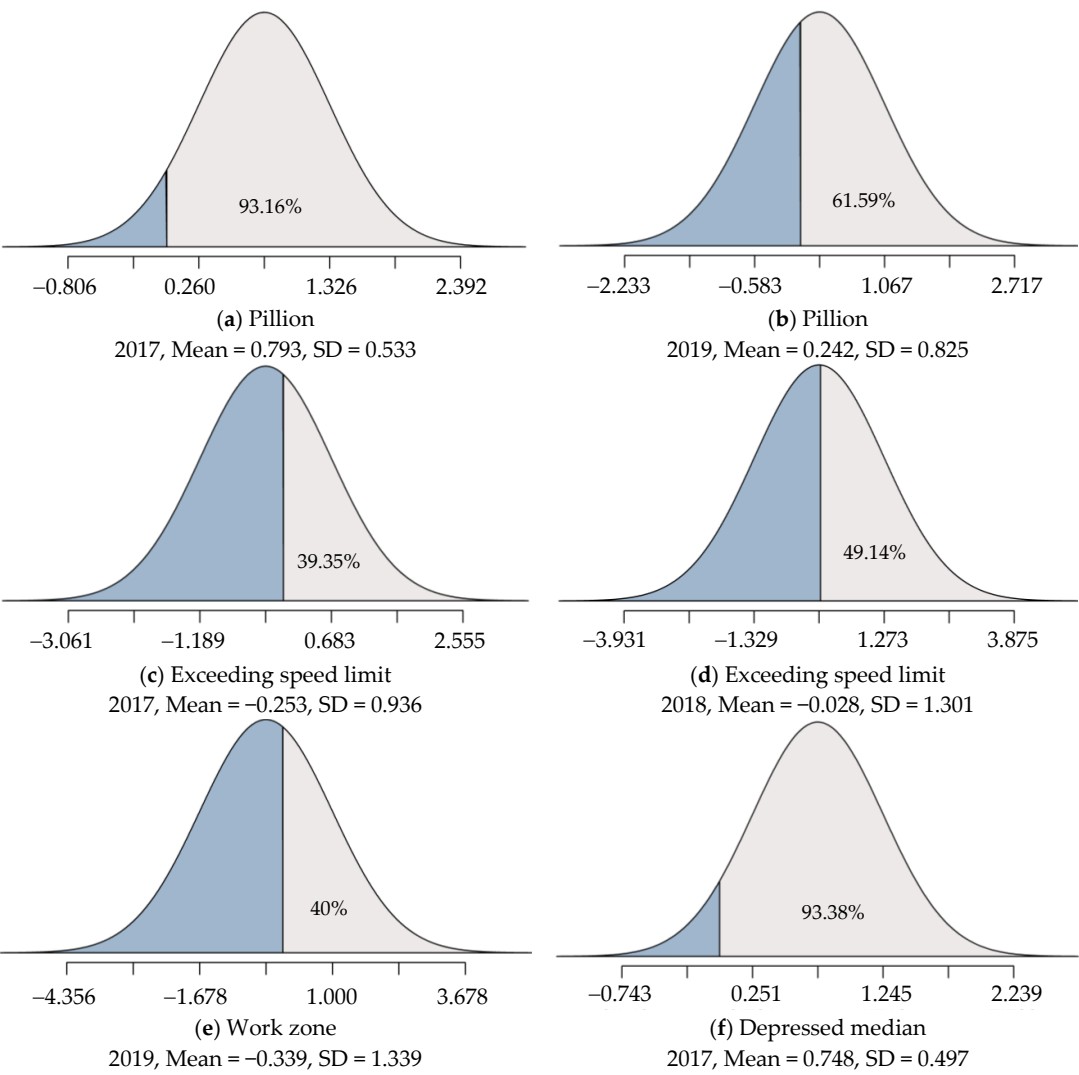

**Figure 4.** *Cont.*

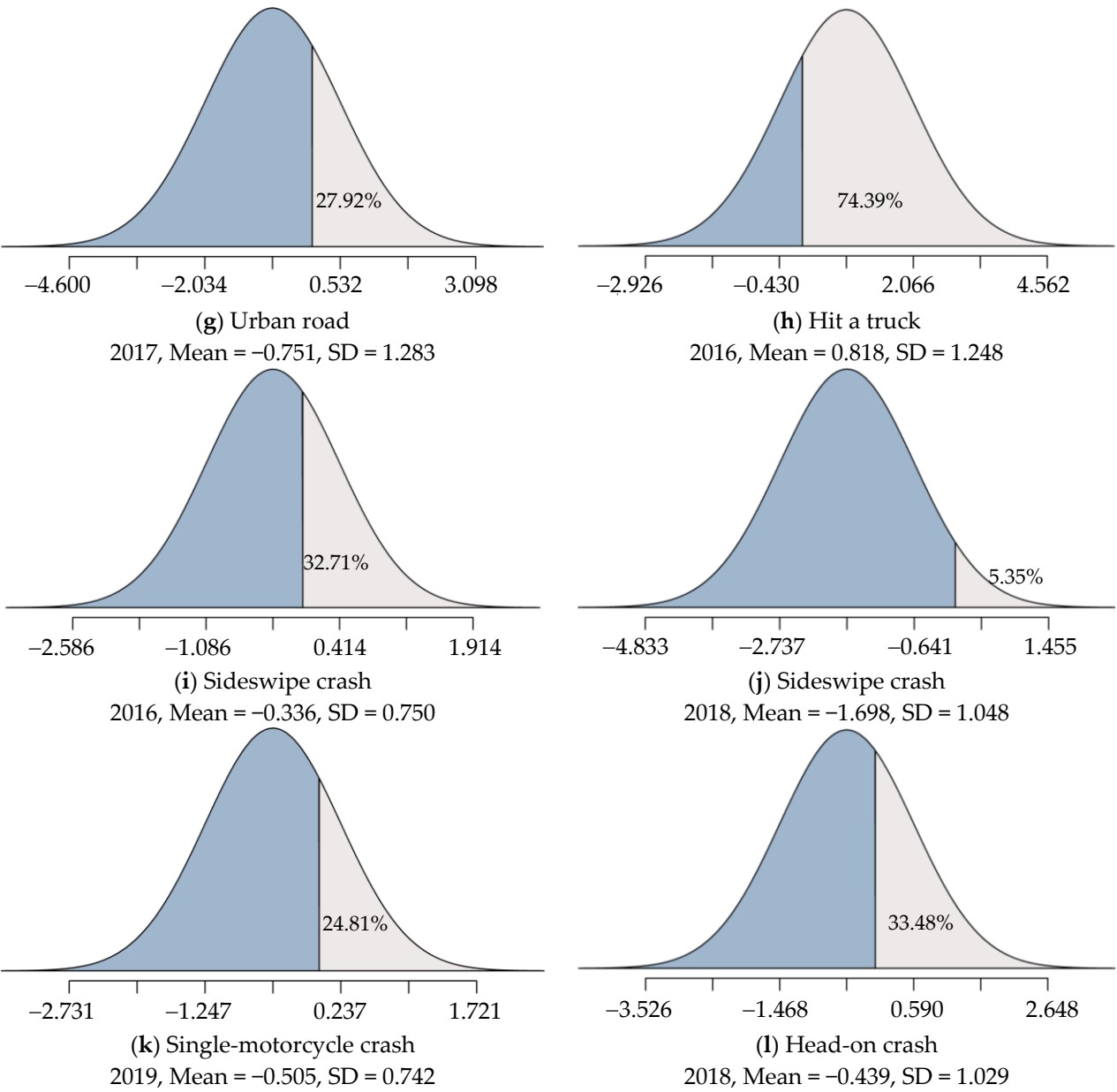

**Figure 4.** Distributional split of the random parameters in daytime crash models.

The variable representing riders exceeding speed limits produced random parameters in the 2017 daytime, 2018 daytime, and 2019 nighttime models, with 39.35% (Figure 4c), 49.14% (Figure 4d), and 65.62% (Figure 5a) of the speeding crashes more likely to cause fatal injuries, respectively. When comparing the marginal effect values, it can be concluded that nighttime speeding collisions had a higher probability of fatalities and severe injuries compared to daytime collisions. In addition to the high crash impact brought on by speeding, one probable explanation may be the poor visibility that causes shorter stopping sight distance that hinders the riders from successfully slowing down before the collision. Earlier studies have supported this finding [17,25,26,46]. In terms of temporal instability, we may observe an increase in the percentage of speeding crashes having a higher risk of death and serious injury from 2017 to 2019, whereas the percentage of speeding crashes with a lower risk of severe or fatal crashes decreased. A possible reason may be that police reporting practices may have changed or improved over time, resulting in more frequent identification of speeding as the cause of serious accidents [16,17,52].

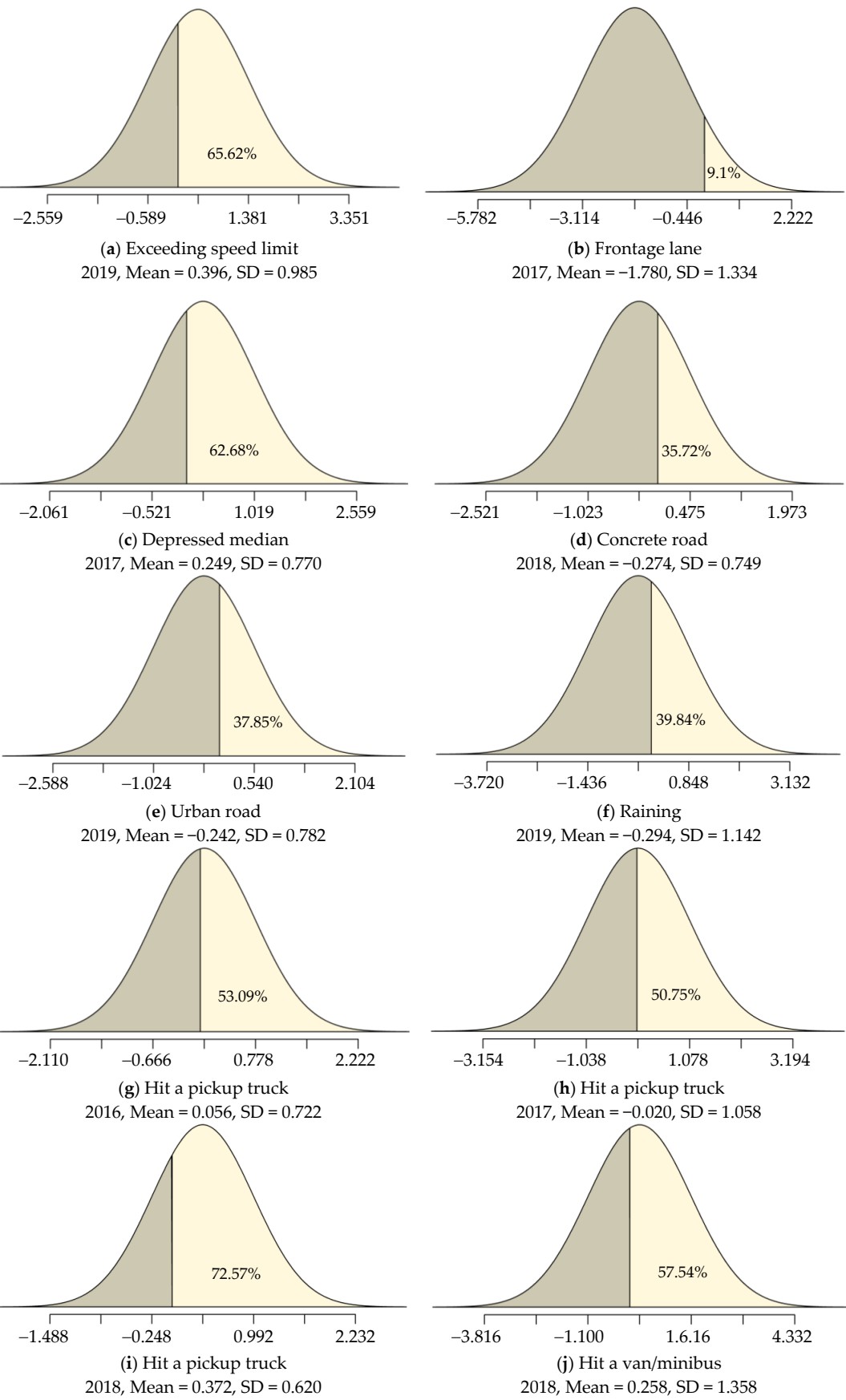

**Figure 5.** *Cont.*

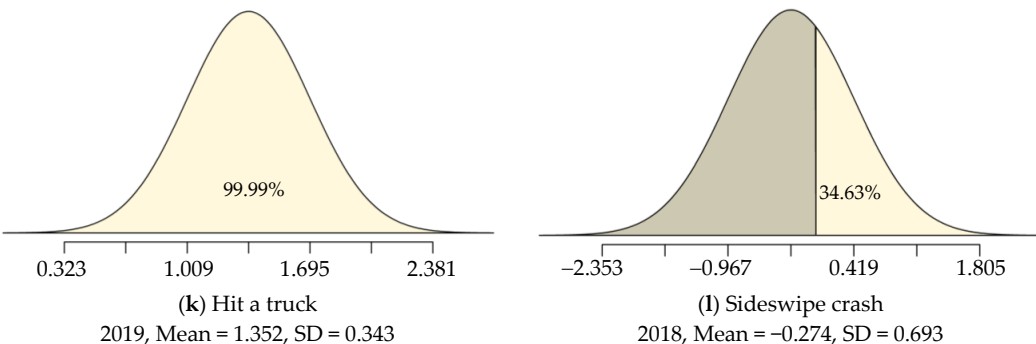

**Figure 5.** Distributional split of the random parameters in nighttime crash models.

The findings demonstrate that riders involved in crashes caused by unauthorized overtaking produced significant characteristics only in 2016, 2017, and 2018 daytime models. When considering the magnitudes of its marginal effect, riders in these accidents had a strikingly high probability of passing away and suffering serious injuries (0.19336, 0.16662, and 0.29066). This finding seems acceptable given that motorcycle overtaking collisions are frequently high-impact collision types including head-on collisions [17,25,26,46]. A possible reason why it was not prominent in the nighttime model may be due to poor visibility, which makes riders more careful when choosing to overtake other vehicles. However, riders may misjudge the relative speed and distance of other vehicles because they overestimate their judgment based on their full vision during the daytime.

Concerning riders' health conditions at the time of the crashes, rider weariness was more likely to cause fatal injuries in every nighttime crash model (apart from 2017), with relatively high probabilities of 0.22736 (2016), 0.42826 (2018), and 0.22972 (2019). This result makes sense given that riders are more likely to be tired and fatigued at nighttime, which may cause a reduction in reaction time, attentiveness, and capacity to constantly regulate motorcycle stability while being directly exposed to external environments [74].

### 6.2. Roadway Characteristics

The model results show that roadway characteristics and crash sites are also highly related to the seriousness of motorcycle injuries.

Riders who were involved in crashes on frontage lanes had a lower risk of suffering severe and fatal injuries both during the daytime and nighttime. Additionally, these effects persisted from 2016 through the 2018 model. The outcome also demonstrates that this variable created random parameters in the 2017 nighttime model, where 90.9% of the nighttime crashes on frontage lanes were likely to cause just minor injuries (Figure 5b). Plausible answers are low-speed local traffic and low traffic volume (less frequency of cars and trucks) in the frontage lane [25].

Regarding work zone locations, especially for the 2019 model, the results indicate that riders engaged in incidents on a road undergoing maintenance or construction were more likely to suffer significant injuries or risk dying in crashes. In addition to creating fixed-effect parameters in the 2019 nighttime model, this variable created random parameters in the 2019 daytime model, with 40% of the work zone collisions being prone to causing fatalities and serious injuries (Figure 4e). Based on these findings, it can be concluded that nighttime work zone crashes carry a greater risk of fatalities and serious injuries compared to daytime incidents. The reason for the reported temporal instability (being insignificant in the 2016–2018 model and significant in the 2019 model) may be that, from time to time, work zone locations may vary from one area to another and are overseen by various construction firms with varying levels of safety implementation (e.g., in 2019, some construction firms may not have implemented adequate safety measures, suggesting a decrease in the implementation of safety practices during that period). A prior study also found that work-zone motorcycle accidents had a higher risk of fatalities and serious injuries compared to non-work-zone crashes [70].

Regarding the number of lanes, the findings show that riders involved in collisions on two- or four-lane roads were more likely to cause fatal injuries during the daytime (2016 and 2019) and nighttime (2016, 2018, and 2019) compared to collisions on roads with six or more lanes. However, based on the size of the marginal effects, it can be seen that crashes on two-lane roads had a much higher risk of death and serious injuries compared to crashes on four-lane roads. This may be because roads with two lanes are frequently undivided and located in rural areas with higher speed limits; as a result, crashes on these roads are prone to high impacts such as head-on collisions and speed-related crashes. Another reasonable explanation is that crashes on rural roads (often having two or four lanes) were generally found to have a higher likelihood of causing serious injury or death compared to those that occur on urban roadways [26,44].

Regarding road median types, the findings show that crashes on flush-median roadways were more likely to cause severe and fatal injury both during daytime (2017–2018) and nighttime (2017–2018). Interestingly, while incidents on raised-median roads had a reduced risk of death and serious injuries in the daytime (2016 and 2019), these crashes were more likely to cause fatal injuries at nighttime (2018). A possible theory may be that because higher-median roadways are typically found in metropolitan areas, nighttime urban riders may be more likely to accelerate up due to lower traffic volume in low-vision conditions at nighttime (daytime traffic is denser than nighttime traffic in urban areas). Additionally, some urban nighttime riders' driving may be impaired, or they may be driving while under the influence of alcohol. The model output shows that motorcycle accidents on depressed-median roads are prone to causing serious and fatal injuries during the daytime (2017–2018) and nighttime (2017). In both the daytime and nighttime 2017 models, this variable generated random parameters, with 93.38% and 62.68%, respectively, of these crashes resulting in a higher probability of fatalities and serious injuries (Figures 4f and 5c). These findings make sense given that depressed-median roads are typically found in rural areas with substantially higher speed restrictions compared to roads in metropolitan areas [75].

Regarding the pavement types when crashes occur, the findings show that riders involved in incidents on concrete roads were less likely to suffer severe and fatal injuries in the daytime (2017–2018) compared to crashes on asphalt pavements. On the other hand, crashes on concrete roadways during nighttime were found to have a higher probability of fatalities and serious injuries compared to asphalt pavements (2016 and 2018 (see Figure 5d)). These results unequivocally show nontransferability between daytime and nighttime collisions. A possible explanation for this is that concrete roads are also typically located in urban areas where there is heavy traffic and low speed restrictions. However, the change in the effect of this variable in nighttime crashes may be attributed to the nature of nighttime urban riders, trip objectives (traveling to/from entertainment centers, etc.), speed selection (higher speed due to fewer vehicles on the road), traffic volume, and visual conditions.

Regarding the alignment of the roads, the findings show that crashes on curving roads were more likely to cause severe and fatal injuries in the daytime (2016). Similar to this, riders engaged in collisions on gradient roads also had a higher risk of death and serious injuries during daytime (2018–2019) and nighttime (2016–2017). One reasonable argument may be the greater difficulty in handling motorcycles in such places [18]. Earlier research [18,43] also supported these results.

In this study, two types of intersections were considered in the investigation, namely, four-leg intersections and three-leg intersections (the dataset does not state whether an intersection is signalized or non-signalized). The research shows that riders engaged in four-leg-junction crashes faced a higher probability of fatalities and severe injuries in the daytime (2018). Interestingly, crashes on three-leg crossroads are more likely to cause just moderate injuries in the daytime (2019) but cause severe and fatal injuries at nighttime (2018). Thus, it can be argued that nighttime motorcycle intersection-related crashes had a greater safety risk relative to daytime crashes. Without considering the time of day,

previous research [44,45] also discovered that intersection-related crashes increased the likelihood of a rider being fatally injured.

The findings demonstrate that collisions at U-turns were more likely to cause fatalities and severe injuries in both daytime crashes (2016 and 2019) and nighttime crashes (2016 and 2019). This may be attributed to riders' exposure to extreme conflicts such as crossing and weaving conflicts with oncoming traffic. Moskal et al. [76] and Chen and Pai [77] have reported similar findings.

Regarding location, riders involved in crashes in urban areas were less likely to suffer serious and fatal injuries during daytime (2016–2019) and nighttime crashes (2017–2019) relative to crashes in rural areas. However, this variable led to random parameters in the 2017 daytime and 2019 nighttime models, with 27.92% and 37.85% of the urban crashes having a greater probability of fatalities and serious injuries, respectively (Figures 4g and 5e). The random parameters may capture the effect of the nighttime urban riders' group, which was previously identified as a high-risk group. However, numerous studies have found that rural motorcycle crashes are more prone to severer crashes than urban crashes [22,24,37,38].

*6.3. Environmental Characteristics*

Regarding the factors connected to surrounding environments when the crash occurred, the outcomes indicate that crashes on wet roads were more likely to cause serious and fatal injuries in the nighttime crash models (2016–2019). Because this variable was not significant in the daytime models, it can be concluded that riding at nighttime on wet roads is much riskier than at daytime. Quddus et al. [22,48] similarly found that regardless of the time of day, motorcycle crashes on wet roads resulted in higher risks of severe injury. Surprisingly, the results suggest that crashes that occurred in the rain were less likely to result in serious and fatal injuries at nighttime (2018–2019). In the 2019 model, this variable generated random parameters, with 39.84% of nighttime crashes during rain increasing the risk of fatalities and serious injuries (Figure 5f). A probable argument is that rainy conditions at nighttime may act as a disincentive to risky behaviors such as speeding, aggressiveness, risky overtaking, etc. Once the rain ceases, riders (both those who continue to ride in the rain at a slow speed and those who halt riding to prevent getting wet) are prone to speeding to make up for the time lost from slowing down or waiting for the rain to stop. This result is consistent with earlier research [44,46].

In this study, the day of the week was also found to be strongly correlated to motorcyclists' resulting injury severities. The findings indicate that riders of motorcycles that crashed on weekends had a higher risk of serious and fatal injuries during the daytime (2016–2019) and a higher risk of fatal injuries at nighttime (2017–2018). Plausible reasons may be the lower roadway exposure that offers the rider more opportunity to choose faster speeds and other dangerous habits that are common among weekend riders such as speeding, impaired riding, and riding under the influence of alcohol [20]. This result is consistent with earlier research [29,43].

The light condition was one of the major factors in raising the severity of motorcyclists' injuries at nighttime across all years. The model findings indicate that crashes on dark roads were more likely to cause fatal injuries at nighttime (2016–2019) compared to lit highways. This result seems logical because, with an optimal lighting configuration, riders will have better sight and longer reaction time [37,78]. Prior studies have supported this result [17,26,43].

*6.4. Crash Characteristics*

Regarding other involved party variables, the data showed crashes involving motorcycles and pickups, vans, buses, or trucks were more likely to result in fatalities and serious injuries both at daytime and at nighttime in comparison with crashes between motorcycles or a motorcycle and passenger vehicles (In some models, the pickup, van/bus, and truck variables also provided normally distributed random parameters, with the majority of the crashes having a higher probability of fatal injuries, as shown in Figures 4h and 5g–k).

A plausible explanation is that collisions against large vehicle types could produce high-impact collision forces that could result in serious and fatal injuries. Other logical explanations may be that the riders are likely to collide with sharp objects or corners of the other vehicles, and motorcyclists' bodies are more immediately exposed to potential harm without energy-dissipating structures and safety features [17]. Similar results were also revealed in previous investigations [30,33,34,36,37,46].

Regarding collision types, the findings show that motorcycle rear-end collisions were less severe and had lower rates of fatal injuries during daytime (2016 and 2018) and nighttime (2019) relative to other crash types. Interestingly, although sideswipe crashes typically resulted in minor injuries across all years of daytime and nighttime crashes, this variable produced random parameters in the 2016 daytime, 2018 daytime, and 2018 nighttime models, with 32.71%, 5.35%, and 34.63% of the sideswipe crashes having a higher risk of death and serious injuries (Figure 4i,j and Figure 5l), respectively. Similarly, the findings also demonstrate that single-vehicle collisions were less likely to cause severe or fatal injuries both at daytime (2016, 2018, and 2019) and nighttime (2019), whereas head-on collisions were more likely to cause severe and fatal injuries both at daytime (2017–2019) and nighttime (2016–2019). The single-motorcycle crash variable also generated random characteristics during the daytime in 2019, with 24.81% of the single-motorcycle crashes likely to cause fatal injuries (Figure 4k). This percentage of collisions may capture single-vehicle collisions which involved contact with permanent objects like trees and poles [22,24]. The head-on collision variable also generated random parameters of daytime crashes in 2018, with 33.48% of the head-on crashes being more likely to cause serious and fatal injuries (Figure 4l). Geedipally et al. [24] have similarly found that single-vehicle, rear-end (same-direction), and angular collisions (including sideswipes) had lower risks of fatalities compared to head-on collisions, which may be due to reduced speed by riders at or near intersections as they adjust for higher perceived risks [22] and less transfer of energy of these crash types when a motorcycle is involved [24].

### 6.5. Unobserved Heterogeneity in Means and Variances

In addition to allowing all coefficients of variables to vary among crashes (represented by random parameters), this study also investigated the impact of all independent factors on the means and rates of variation of the random parameters identified in each model. This will provide information on the interaction effects of a pair of independent factors on the dependent variable of riders' injury severities.

As shown in Tables 5 and 6, in the 2016 daytime model, accidents on flush-median roads increased the mean of hitting a truck, thereby increasing the probability of fatal injury. The results highlight the significance of taking unobserved variability into account. In other words, the variables reflecting crashes on flush-median roads did not create significant parameters when being utilized as a fixed parameter or random parameter; however, they greatly raised the chance of death among riders who hit trucks. In the 2017 daytime model, riders under the influence of alcohol raised the mean of crashes on depressed-median roads, making fatal injuries more likely. In the 2018 daytime model, collisions on two-lane roads raised the means of the speed-related collisions and sideswipe collisions, making fatal injuries more likely, and collisions on four-lane roads similarly increased the mean of sideswipe collisions, increasing the likelihood of a fatal injury. In the 2019 daytime model, crashes on depressed-median roads increased the mean of work-zone crashes, thereby making fatal injuries more likely. In the 2017 nighttime model, riders under the influence of alcohol increased the mean of crashes in frontage lanes, thus making fatal injuries more likely. In the 2018 nighttime model, curved-road crashes raised the probability of hitting a pickup truck, increasing the likelihood of fatal casualties. Last but not least, in the 2019 nighttime model, crashes within U-turn locations increased the mean of crashes during rainy weather, increasing the likelihood of fatal injuries.

Regarding heterogeneity in variance results (Tables 5 and 6), in the 2017 daytime model, crashes on main-lane roads reduced the variation in the effect of speeding and

crashes in urban areas, whereas it increased the variation in the effect of hitting pickup trucks on outcomes of motorcyclist injury severity. In the 2018 daytime model, accidents on wet roads caused more variation in the impacts of speeding-related crashes, crashes with crossing objects, and sideswipe accidents on the severity of motorcyclists' injuries. In the 2019 daytime model, speeding collisions increased the variation in the effects of work-zone and single-motorcycle crashes. In the 2016 nighttime model, rear-end collisions increased the variation in the impact of four-lane collisions. In the 2017 nighttime model, speeding crashes decreased the variation in the effect of depressed-median road crashes and increased the variation in the effect of crashes resulting from hitting pickup trucks. In the 2018 nighttime model, collisions on sunken-median roadways increased the variation in the impacts of collisions on concrete roads and sideswipe collisions; also, they lessened the variation in the impact of collisions resulting from hitting pickup trucks. In the 2019 nighttime model, crashes on frontage lanes decreased the variation in the effect of speeding crashes, whereas the variable representing riders under the influence of alcohol increased the variation in the effect of crashes under rainy weather and decreased the variation in the effect of urban road crashes on the severity of motorcyclists' injuries.

## 7. Discussion of Limitations and Directions for Future Work

While this research presents valuable insights, it is important to acknowledge its limitations. Firstly, the study focused exclusively on crash data from Thailand, which means that the conclusions may not be applicable to other developing countries with varying road infrastructure, regulations surrounding road safety, and cultural perceptions of motorcycle usage [54,79]. To improve the breadth of knowledge on this topic, future research could expand its scope by examining comprehensive crash data from developing nations in other regions such as the Middle East or Africa and comparing it to the findings of the current study. Secondly, when the complete data are categorized into daytime/nighttime and yearly data, the sample size of each becomes comparatively lower than the recommended size for a heterogeneity model [80]. However, compared to recent studies [9,17,66,81], the sample sizes of each sub-dataset in this study are considerably larger. Nonetheless, to achieve robust and reliable results, having a larger sample size would be preferable. Thirdly, as this study utilized data from police reports, the information available is based entirely on the subjective judgment of police officers. This means that errors such as misjudgments regarding a vehicle's movements before a crash or omissions of important details (e.g., in this study, helmet use status is not recorded by police officers) are likely to occur [50]. While advanced heterogeneity models may somewhat alleviate the negative effects of leaving out significant explanatory variables, the estimates from the resulting models will not be as effective in accounting for unobserved heterogeneity as when these variables are included in the model specification [50]. Therefore, datasets containing more detailed crash information and important attributes (such as helmet use) would be more informative when included in the model specification. Furthermore, it would be beneficial for a future study to differentiate between urban and rural riders in addition to separating them by time of day and year, as the willingness to use helmets may vary between these two groups [82,83]. Fourthly, this study focused solely on a four-year timeframe spanning from 2016 to 2019. However, it would be beneficial to examine whether the findings remain consistent over a more extended period, particularly before, during, and after the COVID-19 pandemic lockdown [84]. Fifthly, the proposed methodological approach (i.e., the random parameters ordered with heterogeneity in the means and variances model), despite offering significant versatility in capturing the underlying unobserved heterogeneity, has fixed thresholds that pose limitations for severity analysis, as they do not permit the effect of extreme categories to increase or decrease simultaneously (see example in [85]). Despite the computational challenges, future research may develop a model that includes threshold heterogeneity—where thresholds can vary as a function of exogenous variables and across observations [55]—to determine whether such a model would enhance the model fit significantly. Lastly, while some of the factors in the roadway and crash characteristics

category exhibit a consistent effect across significant models, they are also discovered to be statistically insignificant in some models. Explaining these types of instabilities, especially in this variable group, can be challenging because their features do not change easily over a short period. Thus, these instabilities may be associated with unobservable rider characteristics or actions prior to accidents in that specific year that are unknown to analysts. A causal-inference model using a deep investigation method would be effective in seeking causes for this type of instability [85]. Nonetheless, this model's applicability is limited to a small dataset because a thorough investigation requires a considerable amount of time. Furthermore, the model's predictive ability may not be generalizable, as investigations can be location-specific.

## 8. Conclusions and Recommendations

This study sought to determine variables that affect the degree of injuries resulting from motorcycle accidents in the daytime and nighttime, and it examined how those elements have evolved over time. The study employed a mixed-ordered probit model with heterogeneity in means and variances, using data from motorcycle accidents that occurred between 2016 and 2019 on Thai roads nationwide. Due to this paper examining the differences between factors affecting the severity of injuries at daytime and nighttime as well as the temporal stability of those factors, its findings have profound implications for current safety practices, the allocation of funds for safety improvements, and the prioritization of investments for crash countermeasures by providing evidence-based recommendations for practitioners, decision-makers, and policymakers.

This study discovered several critical characteristics that affect the severity of motorcyclists' injuries resulting from nighttime collisions. For instance, the study discovered that male motorcyclists had a higher risk of death and catastrophic injuries compared to females (significant in the latest period of 2018 and at nighttime in 2019). Similarly to this, riders involved in crashes due to exceeding legal speed limits were more likely to sustain major injuries at nighttime. Additionally, this study discovered that deadly crashes were more probable for riders who had fatigue when the crashes occurred (the effects were also temporally stable). Riders involved in collisions within work-zone areas (areas undergoing repair or construction) were more likely to suffer from fatal injuries. Based on the results for nighttime crashes on elevated-median roads and crashes on concrete roads, it can be concluded that nighttime urban riders were more likely to suffer severe and fatal injuries in crashes. Riders involved in nighttime intersection-related crashes had a greater probability of sustaining fatal injuries in a crash. The study indicates that accidents on wet roads were temporally consistent in having a larger chance of fatalities and serious injuries at nighttime. It also shows that nighttime crashes on dark roads were temporally consistent in having a higher risk of fatalities and severe injuries compared to those on lit roads. According to these findings, special emphasis should be placed on the vulnerable male-rider group through education campaigns and educational interventions to increase their knowledge and awareness about nighttime riding safety. To reduce speeding riders at nighttime in metropolitan areas, strict law enforcement measures such as hefty fines or temporary suspension of driving licenses should be put in place, and repeat offenders should face more severe penalties. Regarding work-zone crashes, road safety audits (especially during the building phase) should be undertaken more regularly to lower the probability of collisions and lessen injury severity outcomes. For instance, the results of the audits should guarantee that the relevant contractors supply enough directed lighting systems on the detours, guardrails, reflective material, clean road surfaces, etc. Regarding lighting conditions, increasing street lighting may lessen the severity of the crashes by assisting riders' impaired visibility in the dark or inclement weather [86].

This study identified one significant factor that affects a safety issue for motorcycle riders in only daytime crash results. Riders involved in a collision caused by unauthorized overtaking of other vehicles had an incredibly high risk of suffering fatal injuries in crashes in the daytime (the effects were also temporally stable). As a result, greater efforts should be

made to enhance motorcycle-riding training programs by emphasizing safe riding practices and knowledge on how to properly change lanes and overtake vehicles.

Finally, this study also identified several important variables that impact the severity of both daytime and nighttime motorcycle crashes. For instance, the results showed that riders with the presence of a pillion passenger were more likely to sustain severe and fatal injuries than lone riders (these effects were temporally stable over the years 2016–2019). The study also discovered that frontage-lane roads were temporally consistent in reducing the severity of motorcycle crash injuries compared to highways without frontage lanes. However, crashes on double-lane roads, crashes on depressed- or flush-median roads, and crashes in rural areas (compared to urban areas) had higher risks of fatalities and serious injuries. Additionally, riders involved in U-turn-related crashes had a higher risk of fatal injuries. The research also revealed that motorcycle accidents on weekends are more likely to result in serious and fatal injuries (effects were temporally stable). When large vehicles (pickups, vans, buses, and trucks) were involved in collisions, riders were more likely to suffer fatal injuries. Regardless of the time of the day, head-on collisions were temporally consistent in causing a higher risk of fatalities and serious injuries compared to single-vehicle crashes and rear-end and sideswipe crashes. Based on these findings, improving motorcycle safety may be challenging due to the vulnerability of motorcyclists. Therefore, substantial attention should be focused on improving the effectiveness of motorcycle training, increasing safety awareness, and strengthening law enforcement for policy implications. Motorcycle crashes in frontage lanes were less likely to cause death or serious injuries because a frontage lane can effectively prevent potential collisions or interactions with larger vehicles on the main lane such as pickup trucks, vans, buses, or heavy trucks, which were found to significantly increase riders' risk of being killed or seriously injured in crashes. Therefore, given that the registered motorcycle remains the highest trend in Thailand (and the Asian region), roadway engineers, safety experts, and decision-makers should take into consideration building exclusive motorcycle lanes which can separate motorcycles from other vehicles, reduce accident exposure, and significantly and effectively improve motorcyclists safety [87,88]. Exclusive motorcycle lanes have been created in various Asian nations including Taiwan, Indonesia, and Malaysia [89]. Malaysia has already seen a significant reduction in motorcycle accidents [90].

**Author Contributions:** Conceptualization, C.S. and V.R.; methodology and data curation, C.S. and T.C.; formal analysis and investigation, C.S., T.C. and P.W.; writing—original draft preparation, C.S.; writing—review and editing, T.C., S.J. and W.L.; funding acquisition, S.J. and V.R.; project administration and supervision, S.J. and V.R. All authors have read and agreed to the published version of the manuscript.

**Funding:** This research was funded by the Suranaree University of Technology under grant number IRD7-704-65-12-35.

**Institutional Review Board Statement:** This research was approved by the Human Research Ethics Committee of Suranaree University of Technology under Code of Approval (COA) No.18/2565 (11 March 2022).

**Informed Consent Statement:** Not applicable.

**Data Availability Statement:** Data will be made available upon reasonable request.

**Acknowledgments:** The authors would like to thank Suranaree University of Technology for supporting this research project through grant number IRD7-704-65-12-35. The author would like to acknowledge the Department of Highways of Thailand for supplying the nationwide road traffic crash data.

**Conflicts of Interest:** The authors declare that they have no known competing financial interests or personal relationships that could have appeared to influence the work reported in this paper.

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
