# Peer review of "Temporal Instability and Transferability Analysis of Daytime and Nighttime Motorcyclist-Injury Severities Considering Unobserved Heterogeneity of Data"

_sustainability, doi:10.3390/su15054486_

Round 1
Reviewer 1 Report
Thank you for giving me this opportunity to read the manuscript entitled " Temporal Instability and Transferability Analysis of Daytime and Nighttime Motorcyclist-Injury Severity Considering Unobserved Heterogeneity of Data". The manuscript uses motorcycle crash data from 2016 to 2019 to investigate the risk factors associated with the severity of motorcyclist injuries sustained in daytime and nighttime motorcycle crashes in Thailand. The study applies mixed-ordered probit models with means and variances heterogeneity to account for unobserved heterogeneity and also explores the temporal instability of risk factors. The topic of this manuscript is interesting and would be a good contribution to this field. I think it could be considered for publication once the following issues are addressed.
1. The study only uses data from Thailand, and the results may not be generalizable to other countries with different road infrastructure, road safety regulations, and cultural attitudes towards motorcycle use.
2. The sample size of the study may be limited, which could impact the robustness of the results.
3. The study relies on self-reported data, which could be subject to biases and missing values.
4. The study only covers a four-year period, and it would be useful to see if the results hold over a longer time period.
5. The use of mixed-ordered probit models with means and variances heterogeneity may have certain limitations in terms of the interpretation of the results and the potential for omitted variable bias.
6. Limitation section should be added as a sub-section to the Discussion.
7. Some grammatical errors exist in the manuscript. Therefore, a critical review of the manuscript's language will improve its readability.
Reviewer 2 Report
This paper determined the risk factors that influence the severity of motorcyclist injuries sustained in the daytime and nighttime motorcycle crash in Thailand using motorcycle crash data from 2016 to 2019. The language can be improved but it is understandable. The introduction is not written following the journal instructions.
In this paper 13795 motorcyclist crashes that occurred from 2016 to 2019 were analysed. The data are not clearly describing the population i.e. what % of the crashes happened in daytime or what is the number of males and females. 38daytime/39 nightime attributes were categorised into four types of variables: rider characteristics, roadway characteristics, environmental characteristics, and crash characteris considering three levels of motorcyclist injury severity.
The autohs state that:
|
In Thailand, rural riders are less likely to wear helmets compared |
631 |
|
to urban counterparts, because metropolitan areas have stricter law enforcement and |
632 |
|
more campaigns that promote helmet-wearing. Jomnonkwao, et al. [77] and Champahom, 633 |
|
|
et al. [78] discovered that encouraging the importance of wearing helmets via television, |
634 |
|
signs, and posters to enlighten people on the benefits of wearing helmets and creating a |
635 |
|
course of action that encourages motorcycle users in rural society to understand the seri- |
636 |
|
ous risks of riding without a helmet could increase helmet use among rural riders. |
This means that the authors understand well that protective equipment has a very large impact on injury severity levels.
Since the data used in this study lack very important information about the protective equipment used by the motorcyclists involved in the crashes this represents the biggest drawback of this paper. The information on protective equipment can drastically change the numbers presented in this paper and therefore the authors should consider this very carefully. Without this data, as the authors stated themselves are different from urban to rural areas the obtained results can be misleading and can lead to wrong interpretations and even wrong countermeasures. Therefore this data must be included in order to provide relevant conclusions and results interpretations.
Reviewer 3 Report
The manuscript proposes mixed-ordered probit models with means and variances heterogeneity based on motorcycle crash data from 2016 to 2019. Using the established model, the study analyzes the differences among the risk factors that influence the severity of motorcyclist injuries sustained in daytime and nighttime motorcycle crashes in Thailand, as well as the temporal stability of those factors. Overall, the findings of this study have certain reference significance for current safety practices and the formulation of relevant policies.
However, I have minor concerns about the methodology and presentation as follows:
1 Line 215: Can the author give the description of [POD]?
2 At the beginning of 6.1, the author mentions that the male riders had a reduced risk of death and serious injuries in 2016, but a higher risk of death and serious injuries in 2018 and 2019 nighttime. Although the author gives the reason for the change in the result of the nighttime crash model in the manuscript, it is not sufficient, e.g., why 2017 is a transition year.
3In Section 6, the author establishes the daytime crash model and the nighttime crash model separately to explore the yearly instability of the risk factors. However, the yearly instability is not explained clearly in the manuscript.
Round 2
Reviewer 2 Report
Dear authors,
I believe that the results presented in this paper can be misleading and can lead to wrong interpretations and even wrong countermeasures in your country. Therefore Im against publication in its present form.
kind regards
